# Agreement-on-the-Line: Predicting the Performance of Neural Networks under Distribution Shift

**Christina Baek**[1]    **Yiding Jiang**[1]    **Aditi Raghunathan**[1]    **Zico Kolter**[1,2]

[1]Carnegie Mellon University, [2]Bosch Center for AI

`{kbaek, yidingji, raditi, zkolter}@cs.cmu.edu`

## Abstract

Recently, Miller et al. [59] showed that a model's in-distribution (ID) accuracy has a strong linear correlation with its out-of-distribution (OOD) accuracy on several OOD benchmarks — a phenomenon they dubbed "accuracy-on-the-line". While a useful tool for model selection (i.e., the models with better ID accuracy are likely to have better OOD accuracy), this fact does not help estimate the *actual* OOD performance of models without access to a labeled OOD validation set. In this paper, we show a similar but surprising phenomenon also holds for the *agreement* between pairs of neural network classifiers: whenever accuracy-on-the-line holds, we observe that the OOD agreement between the predictions of *any* two pairs of neural networks (with potentially different architectures) also observes a strong linear correlation with their ID agreement. Furthermore, we observe that the slope and bias of OOD vs. ID agreement closely matches that of OOD vs. ID accuracy. This phenomenon, which we call *agreement-on-the-line*, has important practical applications: without any labeled data, we can *predict the OOD accuracy of classifiers*, since OOD agreement can be estimated with just unlabeled data. Our prediction algorithm outperforms previous methods both in shifts where agreement-on-the-line holds and, surprisingly, when accuracy is not on the line. This phenomenon also provides new insights into deep neural networks: unlike accuracy-on-the-line, agreement-on-the-line appears to only hold for neural network classifiers.

## 1 Introduction

Machine learning operates well when models observe and make decisions on data coming from the same distribution as the training data. Yet in the real world, this assumption rarely holds. Environments are never fully controlled. Robots interact with their surroundings, effectively changing what they see in the future. Self-driving cars face constant distribution shift when driving to new cities under changing weather conditions. Models trained on clinical data from one hospital face challenges when deployed for a different hospital with different subpopulations. Under these premises, practitioners constantly face the problem of estimating a model's performance on new data distributions (*out-of-distribution*, or OOD) that are related to but different from the data distribution that the model was trained on (*in-distribution*, or ID). Depending on the distribution shift, models may sometimes break catastrophically under new conditions, or may only suffer a small degradation in performance. Differentiating between such cases is crucial in practice.

Assessing OOD performance is difficult because in reality, labeled OOD data is often very costly to obtain. On the other hand, *unlabeled* OOD data is much easier to obtain. A natural question is whether we can leverage *unlabeled* OOD data for estimating the OOD performance. This paradigm of using unlabeled data to predict the OOD generalization performance has received much attention recently [29, 9, 87, 22, 23, 10, 31]. Although many different metrics have been proposed, their success varies widely depending on the shift and the ID performance of the model. While it is

36th Conference on Neural Information Processing Systems (NeurIPS 2022).

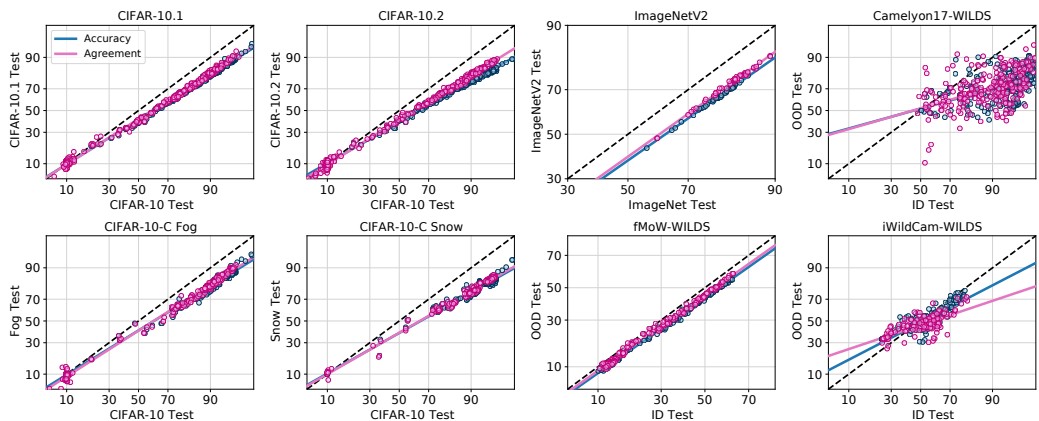

**Figure 1:** When ID vs. OOD accuracy is linearly correlated, the ID vs. OOD agreement is also linearly correlated. Additionally, when ID vs. OOD accuracy is not linearly correlated, agreement is also not linearly correlated. Each blue point in the scatter plot represents the accuracy of a single model. Each pink point represents the agreement between a pair of models. To avoid cluttering the figure, given $n$ models of interest, we only plot $n$ random pairs. The axes are probit scaled as described in Section 3.2.

impossible for a method to always work with no assumptions [29], a major obstacle to using these methods is that there is currently no understanding of when they work or a recipe to detect when their predictions might be unreliable.

In a separate but related line of work, Miller et al. [59] demonstrated that in a wide variety of common OOD prediction benchmarks such as CIFAR-10.1 [67], ImageNetV2 [67], CIFAR-10C [34], fMoW-WILDS [13], there exists an almost perfect positive linear correlation between the ID test vs. OOD accuracy of models. When this phenomenon, called *accuracy-on-the-line*, occurs, improving performance on the ID test data directly leads to improvements on OOD performance. Furthermore, if we have access to the slope and bias of this correlation, predicting OOD accuracy becomes straightforward. Unfortunately, accuracy-on-the-line is not a universal phenomenon. In some datasets, such as Camelyon17-WILDS [1], models with the same ID test performance have OOD performance that varied largely. Thus, while the accuracy-on-the-line phenomenon is interesting, its practical use is somewhat limited since determining whether accuracy-on-the-line holds requires labeled OOD data in the first place.

In this work, we begin by observing an analogous phenomenon based upon *agreement* rather than accuracy. Specifically, if we consider pairs of neural network of classifiers, and look at the agreement of their predictions (the proportion of cases where they make the same prediction, which requires no labeled data to compute), we find that there *also* often exists a strong linear correlation between ID vs. OOD agreement. We call this phenomenon *agreement-on-the-line*. Importantly, however, this phenomenon appears to be tightly coupled with accuracy-on-the-line: when agreement-on-the-line holds, accuracy-on-the-line also holds; and when agreement-on-the-line *does not* hold, neither does accuracy-on-the-line. Furthermore, when these properties hold, the linear correlations of both accuracy-on-the-line and agreement-on-the-line appear to have roughly *the same slope and bias*. Interestingly, unlike accuracy-on-the-line, which appears to be a general phenomenon, agreement-on-the-line, especially the fact that the slope and bias of the linear correlation agree across accuracy and agreement, appears to occur only for neural networks. Indeed, the phenomenon is quite unintuitive, given that there is no a-priori reason to believe that agreement and accuracy would be connected in such a manner; nonetheless, we find this phenomenon occurs repeatedly across multiple datasets and vastly different neural network architectures.

This phenomenon is of immediate practical interest. Since agreement-on-the-line can be validated *without* any labeled OOD data, we can use it as a proxy to assess whether accuracy-on-the-line holds, and thus whether ID accuracy is a reasonable OOD model selection criteria. Furthermore, since the slope and bias of the agreement-on-the-line fit can *also* be estimated without labeled OOD data, (for the cases where agreement-on-the-line holds) we can use this approach to derive a simple algorithm for estimating the OOD generalization of classifiers, without *any* access to labeled OOD data. The approach outperforms competing methods and predicts OOD test error with a mean

absolute estimation error of $\leq 2\%$ on datasets where agreement-on-the-line holds. On datasets where agreement-on-the-line does not hold, the method as expected does not perform as well, but surprisingly *still* outperforms competing methods in terms of predicting OOD performance.

To summarize, our contributions are as follows:

1. We discover and empirically analyze the agreement-on-the-line phenomenon: that ID vs. OOD *agreement* for pairs of neural network classifiers lies on a line precisely when the corresponding ID vs. OOD *accuracy* also lies on a line. Furthermore, the slope and bias of these two lines are approximately equal.

2. Leveraging this phenomenon, we develop a simple method for estimating the OOD performance of classifiers without *any* access to labeled OOD data (and by observing whether agreement-on-the-line holds, the method also provides a "sanity check" that these estimates are reasonable). The proposed method outperforms all competing baselines for this task.

## 2   Related Works

**Accuracy-on-the-line**    Before Miller et al. [59], several other works proposing new benchmarks for performance evaluation [58, 67, 69, 85] have also observed a strong linear correlation between ID and OOD performance of models. Theoretical analysis of this phenomenon is limited. Miller et al. [59] constructs distribution shifts where accuracy-on-the-line can be observed assuming the data is Gaussian, however they do not incorporate any assumptions about the classifier. Another work by Mania and Sra [55] analyzes the phenomenon under the assumption that given two models, it is unlikely that the lower accuracy model classifies a data point correctly while the higher accuracy model classifies it incorrectly. Such model similarity has been observed by [56, 30]. Recent works have also observed some nonlinear correlations [47] and negative correlations in performance [79, 41]. In this work, we focus on the results from Miller et al. [59] and leave these nuances for future study.

**Estimating ID generalization via agreement.**    Departing from conventional approaches based on uniform convergence [63, 26, 3, 60], several recent works [62, 89, 28, 39] propose different approaches for estimating generalization error. In particular, this work is closely related to Jiang et al. [39], which shows that the disagreement between two models trained with different random seeds closely tracks the ID generalization error of the models, if the ensemble of the models are well-calibrated. Predicting ID generalization via disagreement has also previously been proposed by Madani et al. [54] and Nakkiran and Bansal [61]. Our method also uses disagreement for estimating performance but, unlike these works, we focus on OOD generalization, and, more importantly, we do not require calibration or models with the same architecture.

**OOD generalization.**    The problem of characterizing generalization in the OOD setting is even more challenging than the ID setting. Ben-David et al. [5] provides one of the first uniform-convergence-based bounds for *domain adaptation*, a related but harder framework of improving the OOD performance of models given unlabeled OOD data and labeled ID data. Several works [57, 18, 46] build on this approach and extend it to other learning scenarios. Most of these works attempt to bound the difference between ID and OOD performance via a certain notion of closeness between the original distribution and shifted distribution (e.g., the total variation distance and the $\mathcal{H}\Delta\mathcal{H}$ divergence which is related to agreement), and build on the uniform-convergence framework [68]. As pointed out by Miller et al. [59], these approaches provide upper bounds on the OOD performance that grows looser as the distribution shift becomes larger, and the bounds do not capture the precise trends observed in practice. Predicting the actual OOD performance using unlabeled data has gained interest in the past decade. These methods can roughly be divided into three categories:

**1. Placing assumptions on the distribution shift.**    Donmez et al. [24] assume knowledge of the marginal of the shifted label distribution $P(y)$ and show that OOD accuracy can be predicted if the shifted distribution satisfies several properties. Steinhardt and Liang [74] work under the assumption that the data $x$ can be separated into "views" that are conditionally independent given label $y$. Chen et al. [10] assume prior knowledge about the shift and use an importance weighting procedure.

**2. Placing assumptions on the classifiers.**    Given multiple classifiers of interest, Platanios et al. [64, 65] form logical constraints based on assumptions about the hypothesis distribution to identify

| Dataset | Accuracy | | | Agreement | | | Confidence Interval |
|---|---|---|---|---|---|---|---|
| | Slope | Bias | $R^2$ | Slope | Bias | $R^2$ | |
| CIFAR-10.1v6 | 0.842 | -0.216 | 0.999 | 0.857 | -0.205 | 0.997 | (-0.046, 0.017) |
| CIFAR-10.2 | 0.768 | -0.287 | 0.999 | 0.839 | -0.226 | 0.996 | (-0.120, -0.030) |
| ImageNetv2 | 0.946 | -0.309 | 0.997 | 0.972 | -0.274 | 0.993 | (-0.0720, 0.061) |
| CIFAR-10C-Fog | 0.834 | -0.228 | 0.995 | 0.870 | -0.239 | 0.996 | (-0.077, 0.053) |
| CIFAR-10C-Snow | 0.762 | -0.289 | 0.974 | 0.766 | -0.266 | 0.974 | (-0.067, 0.047) |
| fMoW-WILDS | 0.952 | -0.163 | 0.998 | 0.954 | -0.121 | 0.995 | (-0.042, 0.030) |
| Camelyon17-WILDS | 0.373 | 0.046 | 0.263 | 0.381 | 0.075 | 0.226 | - |
| iWildCam-WILDS | 0.700 | -0.037 | 0.738 | 0.411 | -0.094 | 0.424 | - |

**Table 1:** Slope, bias, and coefficients of determination ($R^2$) values of linear correlations between ID vs. OOD accuracy and ID vs. OOD agreement. The slope/bias of these linear correlations match when the $R^2$ value is high (i.e. strong linear correlation). We also look at the 95% confidence interval of the differences in slope for the datasets where we observe strong correlation and are interested in whether the slopes are similar. See Section 3.3 for experimental details.

individual classifier's error. On the other hand, Jaffe et al. [38] relates accuracy to the classifiers' covariance matrix under the assumption that classifiers make independent errors and do better than random.

**3. Empirically measuring the distribution shift.** A group of works [27, 71, 22, 23] train a regression model over metrics that measure the severity of the distribution shift. Inspired by the observation that the maximum softmax probability (or confidence) for OOD points is typically lower [35, 34], Guillory et al. [31] and more recently Garg et al. [29] utilize model confidence to predict accuracy. Chuang et al. [14] uses agreement with a set of domain-invariant predictors as a proxy for the unknown, true target labels. This method was extended upon by Chen et al. [9] which improves the predictors by self-training. Yu et al. [87] observed that the distance between the model of interest $f$ and a reference model trained on the pseudolabels of $f$ showed strong linear correlation with OOD accuracy.

Though a large number of methods have been proposed, for the large majority, it is not well-understood when they will work. Intuitively, no method will work on all shifts without additional assumptions [29]. But is there some *simple general structure* to shifts in the real world that allows us to reliably predict OOD accuracy? Even if such a structure is not universal, can we easily *inspect* if this structure holds? What is a plausible assumption we can make about the OOD *behaviour of classifiers*? The novelty and significance of our work comes from trying to better understand and address these questions, specifically for neural networks. In this work, we observe a phenomenon related to, but stronger than accuracy-on-the-line that allows us to reliably predict the OOD accuracy of neural networks.

## 3 The agreement-on-the-line phenomenon

### 3.1 Notation and setup

Let $\mathcal{H}$ denote a set of neural networks trained on $(X_{\text{train}}, \boldsymbol{y}_{\text{train}}) = \{(x_i, y_i)\}_{i=1}^{m_{\text{train}}}$ sampled from $\mathcal{D}_{\text{ID}}$.

Given any pair of models $h, h' \in \mathcal{H}$, for a distribution $\mathcal{D}$, the expected accuracy and agreement are defined as:

$$\text{Acc}(h) = \mathbb{E}_{x,y\sim\mathcal{D}}[\mathbb{1}\{h(x) = y\}], \quad \text{Agr}(h, h') = \mathbb{E}_{x\sim\mathcal{D}}[\mathbb{1}\{h(x) = h'(x)\}]. \quad (1)$$

We assume access to a labeled validation set $(X_{\text{val}}, \boldsymbol{y}_{\text{val}}) = \{(x_i, y_i)\}_{i=1}^{m_{\text{val}}}$ sampled from $\mathcal{D}_{\text{ID}}$ that allows us to estimate the ID accuracy $\widehat{\text{Acc}}_{\text{ID}}(h)$ as the sample average of $\mathbb{1}\{h(x) = y\}$ over the validation set. We do not assume access to a labeled OOD validation set, as this is often impractical to obtain, and thereby cannot directly estimate $\widehat{\text{Acc}}_{\text{OOD}}(h)$ in a similar manner.

Agreement, on the other hand, only requires access to unlabeled data. We assume access to *unlabeled* samples $X_{\text{OOD}} = \{x_i\}_{i=1}^{m_{\text{OOD}}}$ from the shifted distribution of interest $\mathcal{D}_{\text{OOD}}$. Hence, we can estimate

both the ID and OOD agreement as follows:

$$\widehat{\mathrm{Agr}}_{\mathsf{ID}}(h, h') = \frac{1}{m_{\mathsf{val}}} \sum_{x \in X_{\mathsf{val}}} \mathbb{1}\{h(x) = h'(x)\}, \quad \widehat{\mathrm{Agr}}_{\mathsf{OOD}}(h, h') = \frac{1}{m_{\mathsf{OOD}}} \sum_{x \in X_{\mathsf{OOD}}} \mathbb{1}\{h(x) = h'(x)\} \quad (2)$$

## 3.2 Experimental setup

We study the ID vs. OOD accuracy and agreement between pairs of models across more than 20 common OOD benchmarks and hundreds of independently trained neural networks.

**Datasets.** We present results on 8 dataset shifts in the main paper, and include results for other distribution shifts in the Appendix C. These 8 datasets span:

1. Dataset reproductions: CIFAR-10.1 [67], CIFAR-10.2 [52] reproductions of CIFAR-10 [43] and ImageNetV2 [67] reproduction of ImageNet [21]
2. Synthetic corruptions: CIFAR-10C Fog and CIFAR-10C Snow [34]
3. Real-world shifts from [42]: satellite images (fMoW-WILDS), images from camera traps in the wildlife (iWildCam-WILDS [4]), and images of cancer tissue (Camelyon17-WILDS [1])

Appendix C includes results on CINIC-10 [19], STL-10 [16], and other WILDS and CIFAR-10C benchmarks. We also investigate an analogous phenomenon for the F1-score used to assess the reading comprehension performance of language models on the Amazon-SQuAD benchmark [58].

**Models.** For ImageNetV2, we evaluate 50 ImageNet pretrained models from the timm [82] package. On the remaining 7 shifts, we evaluate on all independently trained models in the testbed created and utilized by [59] consisting of $\geq 150$ models for each shift. The evaluated models span a variety of convolutional neural networks (e.g. ResNet [33], DenseNet [36], EfficientNet [78], VGG [49]) as well as various Vision Transformers [25]. All architectures and models are listed in the Appendix D.

**Probit scaling.** Miller et al. [59] report their results after probit scaling $\Phi^{-1}(\cdot)$ [1] the ID vs. OOD accuracies due to a better linear fit. We apply the same probit transform to both accuracy and agreement in our experiments.

## 3.3 Observations

We empirically observe a peculiar phenomenon in deep neural networks, which we call *agreement-on-the-line* characterized by the following three properties:

**Prop(i)** When ID vs. OOD accuracy observes a strong linear correlation ($\geq 0.95$ $R^2$ values), we see that ID vs. OOD *agreement is also strongly linearly correlated*.

**Prop(ii)** When both accuracy and agreement observe strong linear correlation, we see that these linear correlations have almost the *same slope and bias*.

**Prop(iii)** When the linear correlation of ID vs. OOD accuracy is weak ($\leq 0.75$ $R^2$ values), the linear correlation between ID and OOD agreement is similarly weak. [2]

We show the agreement-on-the-line phenomenon on 8 datasets in Figure 1 and Table 1. On CIFAR-10.1, CIFAR-10.2, ImageNetV2, CIFAR-10C Fog/Snow, and fMoW-WILDS, we find that both ID vs. OOD accuracy and agreement observe strong linear correlations, and the linear fits have the same slope and bias (Prop(i), Prop(ii)). On the other hand, on the datasets Camelyon17-WILDS and iWildCam-WILDS where accuracy is not linearly correlated, agreement is also not linearly correlated (Prop(iii)).

To ensure that the differences in the slopes is not statistically significant, we construct the following hypothesis test. For each dataset, we randomly sample 1000 subsets of 10 models from the model

---

[1] The probit transform is the inverse of the cumulative density function of the standard Gaussian distribution.

[2] The $R^2$ thresholds were only chosen to discretize the strength of the linear correlations as strong or weak for the 8 shifts. As shown in Appendix C, the phenomenon actually follows a gradient i.e. when the $R^2$ value is higher, the slope/bias of ID vs. OOD accuracy and ID vs. OOD agreement become closer to each other.

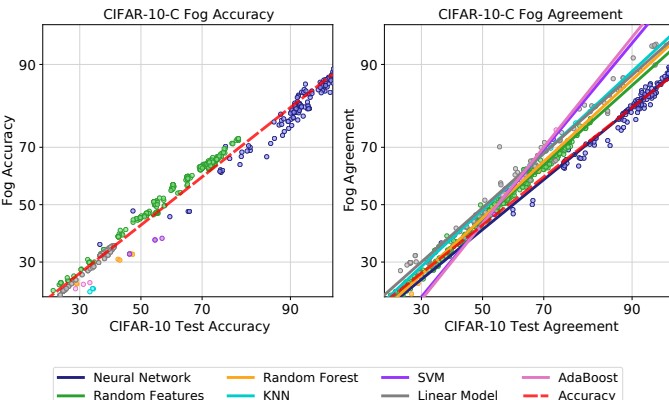

**Figure 2:** We observe whether the agreement-on-the-line phenomenon happens across various model classes on the CIFAR-10 Fog dataset. As shown on the left, the ID vs. OOD accuracy of all model classes lie on the same line. We plot ID vs. OOD agreement between pairs of models from the same model class and observe that only the linear correlation between ID vs. OOD agreement of neural networks match that of ID vs. OOD accuracy (in red).

testbed and compute the corresponding difference in the slope of the linear fits: $\text{Slope}_{\text{Acc}} - \text{Slope}_{\text{Agr}}$. In Table 1, we look at the 95% confidence interval of the distribution of these slope differences for the 6 datasets where we observe a strong correlation. For all datasets with the exception of CIFAR10.2, the null hypothesis (i.e. difference in slope is 0) lies within the 95% confidence interval thus cannot be rejected.

### 3.4 What makes agreement-on-the-line interesting?

First, *agreement can be estimated with just unlabeled data*. Hence, the agreement-on-the-line phenomenon has important practical implications for both *checking* whether the distribution shift observes accuracy-on-the-line and *predicting* the actual value of OOD accuracy without any OOD labels. We present a method to estimate OOD error using this phenomenon in Section 4.

Second, agreement-on-the-line does *not* directly follow from accuracy-on-the-line. Prior work has observed that expected ID accuracy often equals ID agreement over pairs of models with the same architecture, trained on the same dataset but with different random seeds [39]. Agreement-on-the-line goes beyond these results in two ways: (i) agreement between models with *different architectures* (Fig. 2) and (ii) agreement between different checkpoints on the *same training run* (Fig. 4) is also on the ID vs. OOD agreement line. These ID/OOD agreements *do not equal* the expected ID/OOD accuracy. Indeed, understanding why agreement-on-the-line holds requires going beyond the theoretical conditions presented in the prior work [39] which do not hold for this expanded set of models. Furthermore, the phenomenon of matching slopes is *not* due to the models performing well ID and OOD (Accuracy can be thought of as the agreement with the perfect classifier). Even pairs of bad models display this phenomenon. See Appendix F for further discussion.

Finally, we emphasize that there is something special about neural networks that makes the ID vs. OOD agreement trend identical to the ID vs. OOD accuracy trend. This is unlike accuracy-on-the-line that holds across a wide range of models including neural networks and classical approaches. Figure 2 shows CIFAR-10 Test vs. CIFAR-10C Fog accuracy and agreement of linear models (e.g. logistic and ridge regression) and various non-linear models (e.g. Kernel SVM [17], k-Nearest Neighbors, Random Forests [6], Random Features [15], AdaBoost [90]). See plots for other datasets in Appendix C. We look at agreement between pairs of models from the same model family. While Prop(i) seems to hold for several other model families on several shifts, Prop(ii) only holds for neural networks, i.e. the slope and bias of the agreement line *do not match* the slope and bias of the accuracy line for other model families.

## 4 A method for estimating OOD accuracy

In this section, we describe how the phenomenon of agreement-on-the-line (described in Section 3) offers a simple practical method to perform model selection and estimate accuracy under distribution shifts. Recall from Section 3.1 that we have labeled ID validation data $(X_{\text{val}}, \boldsymbol{y}_{\text{val}})$ and unlabeled OOD data $X_{\text{OOD}}$.

**Model selection.** Without OOD labeled data, can we determine which model is likely to achieve the best OOD performance? When accuracy-on-the-line holds and ID vs. OOD accuracy is linearly correlated, we can simply pick the model with highest ID accuracy. In practice, how does one determine if accuracy-on-the-line holds without labeled OOD data? By Prop(i) and Prop(iii), agreement-on-the-line implies accuracy-on-the-line. Hence, we simply need to check if ID and OOD agreement (which can be estimated as in (2)) are linearly correlated, in order to know if our model selection criterion based on ID accuracy is valid.

**OOD error prediction.** Agreement-on-the-line allows us to go beyond model selection and actually *predict OOD accuracy*. Intuitively, we can estimate the slope and bias of the agreement line with just unlabeled data. By Prop(ii), they match the slope and bias of the accuracy line and hence, we can estimate the OOD accuracy by linearly transforming the ID accuracy (with the appropriate probit scaling). We formalize this intuition below and provide an algorithm for OOD accuracy estimation in Algorithm 1. Implementation of our method is available at `https://github.com/kebaek/agreement-on-the-line`.

Recall (Section 3.1) that given $n$ distinct models of interest $\mathcal{H} = \{h_i\}_{i=1}^n$, we can estimate $\mathsf{Acc}_{\mathsf{ID}}(h)$, $\mathsf{Agr}_{\mathsf{ID}}(h, h')$ and $\mathsf{Agr}_{\mathsf{OOD}}(h, h')$ as sample averages over ID labeled validation data and OOD unlabeled data for all $h, h' \in \mathcal{H}$. We now describe an estimator $\widehat{\mathsf{Acc}}_{\mathsf{OOD}}(h)$ for the OOD accuracy of a model $h \in \mathcal{H}$.

From agreement-on-the-line, we know that when ID vs. OOD agreement lies on a line for all $h, h' \in \mathcal{H} \times \mathcal{H}$, ID vs. OOD accuracy for all $h \in \mathcal{H}$ would approximately also lie on the same line:

$$\Phi^{-1}(\mathsf{Acc}_{\mathsf{OOD}}(h)) = a \cdot \Phi^{-1}(\mathsf{Acc}_{\mathsf{ID}}(h)) + b \Leftrightarrow \Phi^{-1}(\mathsf{Agr}_{\mathsf{OOD}}(h, h')) = a \cdot \Phi^{-1}(\mathsf{Agr}_{\mathsf{ID}}(h, h')) + b$$
(3)

We estimate the slope and bias of the linear fit by performing linear regression after applying a probit transform on the agreements as follows.

$$\hat{a}, \hat{b} = \arg\min_{a, b \in \mathbb{R}} \sum_{i \neq j} (\Phi^{-1}(\widehat{\mathsf{Agr}}_{\mathsf{OOD}}(h_i, h_j)) - a \cdot \Phi^{-1}(\widehat{\mathsf{Agr}}_{\mathsf{ID}}(h_i, h_j)) - b)^2$$
(4)

For each model $h \in \mathcal{H}$, given its ID validation accuracy, one could simply plug the estimated slope $\hat{a}$ and bias $\hat{b}$ from (4), and $\widehat{\mathsf{Acc}}_{\mathsf{ID}}(h)$ (sample average over validation set) into (3) to get an estimate of the model's OOD accuracy. We call this *simple* algorithm ALine-S.

Notice that ALine-S does not directly use the OOD agreement estimates concerning the model of interest—we only use agreements indirectly via the estimates $\hat{a}$ and $\hat{b}$. We find that a better estimator can be obtained by *directly* using the model's OOD agreement estimates via simple algebra as follows.

First, note that for any pair of models $h, h' \in \mathcal{H}$, it directly follows from (3) that

$$\frac{\Phi^{-1}(\mathsf{Acc}_{\mathsf{OOD}}(h)) + \Phi^{-1}(\mathsf{Acc}_{\mathsf{OOD}}(h'))}{2} = a \cdot \frac{\Phi^{-1}(\mathsf{Acc}_{\mathsf{ID}}(h)) + \Phi^{-1}(\mathsf{Acc}_{\mathsf{ID}}(h'))}{2} + b \quad (5)$$

By substituting $b = \Phi^{-1}(\mathsf{Agr}_{\mathsf{OOD}}(h, h')) - a \cdot \Phi^{-1}(\mathsf{Agr}_{\mathsf{ID}}(h, h'))$ into (5), we can get that average OOD accuracy of any pair of models $h, h' \in \mathcal{H}$ is

$$\frac{1}{2} \underbrace{\Phi^{-1}(\mathsf{Acc}_{\mathsf{OOD}}(h))}_{\text{unknown}} + \frac{1}{2} \underbrace{\Phi^{-1}(\mathsf{Acc}_{\mathsf{OOD}}(h'))}_{\text{unknown}}$$

$$= \underbrace{\Phi^{-1}(\mathsf{Agr}_{\mathsf{OOD}}(h, h')) + a \cdot \left( \frac{\Phi^{-1}(\mathsf{Acc}_{\mathsf{ID}}(h)) + \Phi^{-1}(\mathsf{Acc}_{\mathsf{ID}}(h'))}{2} - \Phi^{-1}(\mathsf{Agr}_{\mathsf{ID}}(h, h')) \right)}_{\text{known (can estimate via sample average over } X_{\mathsf{OOD}} \text{ and } (X_{\mathsf{val}}, y_{\mathsf{val}}))}. \quad (6)$$

We can plug in estimates of the terms on the right hand side ($\hat{a}$ from linear regression (4)) and the rest from sample averages. In this way, we can construct a system of linear equations of the form (6) involving "unknown" estimates of the probit transformed OOD accuracy of models and other "known" quantities. We solve the system via linear regression to obtain the unknown estimates. We call this procedure ALine-D, and it is described more explicitly in Algorithm 1. Note that there must be at least 3 models in the set of interest $\mathcal{H}$ for the system of linear equations in (6) to have a unique solution.

---
**Algorithm 1** ALine-D: Predicting OOD Accuracy
---
1: **Input:** $m_{\mathrm{ID}}$ validation samples $(X_{\mathrm{ID-val}}, \mathbf{y}_{\mathrm{ID-val}})$, $m_{\mathrm{OOD}}$ unlabeled samples $X_{\mathrm{OOD}}$, a set containing $n$ models of interest $\mathcal{H}$
2: Get $\widehat{\mathrm{Acc}}_{\mathrm{ID}}(h_i) \; \forall i \in [n]$
3: Get $\widehat{\mathrm{Agr}}_{\mathrm{ID}}(h_i, h_j)$ and $\widehat{\mathrm{Agr}}_{\mathrm{OOD}}(h_i, h_j)$ for all pairs of models $i \neq j$
4: Get $\hat{a}, \hat{b} = \arg\min_{a,b \in \mathbb{R}} \sum_{i \neq j} (\Phi^{-1}(\widehat{\mathrm{Agr}}_{\mathrm{OOD}}(h_i, h_j)) - a \cdot \Phi^{-1}(\widehat{\mathrm{Agr}}_{\mathrm{ID}}(h_i, h_j)) - b)^2$
5: Initialize $A \in \mathbb{R}^{\frac{n(n-1)}{2} \times n}$, $\boldsymbol{b} \in \mathbb{R}^{\frac{n(n-1)}{2}}$
6: $i = 0$
7: **for** $h_j, h_k \in \mathcal{H}$ **do**
8: $\quad A_{ij} = \frac{1}{2}, A_{ik} = \frac{1}{2}, A_{i\ell} = 0 \; \forall l \notin \{j, k\}$
9: $\quad \boldsymbol{b}_i = \Phi^{-1}(\widehat{\mathrm{Agr}}_{\mathrm{OOD}}(h_j, h_k)) + \hat{a} \cdot \left( \frac{\Phi^{-1}(\widehat{\mathrm{Acc}}_{\mathrm{ID}}(h_j) + \Phi^{-1}(\widehat{\mathrm{Acc}}_{\mathrm{ID}}(h_k)))}{2} - \Phi^{-1}(\widehat{\mathrm{Agr}}_{\mathrm{ID}}(h_j, h_k)) \right)$
10: $\quad i = i + 1$
11: **end for**
12: Get $\boldsymbol{w}^* = \arg\min_{\boldsymbol{w} \in \mathbb{R}^n} \|A\boldsymbol{w} - \boldsymbol{b}\|_2^2$
13: **return** $\Phi(w_i^*) \; \forall i \in [n]$
---

## 5 Experiments

**Datasets and models.** We evaluate our methods, the simple plug in of slope/bias estimate ALine-S and the more involved ALine-D, on the same models and datasets from Section 3 and two additional datasets CIFAR-10C-Saturate and RxRx1-WILDS (See Appendix C and D for details on these datasets). Specifically, we look at CIFAR-10.1, CIFAR-10.2, ImageNetV2, CIFAR-10C, fMoW-WILDS, and RxRx1-WILDS, where we observe a strong correlation. We also look at the performance of models on datasets where we do not see a strong linear correlation, specifically Camelyon-WILDS and iWildCam-WILDS.

**Baseline methods.** We choose 4 existing unlabeled estimation methods for comparison: Average Threshold Confidence (ATC) by Garg et al. [29], DOC-Feat in Guillory et al. [31], Average Confidence (AC) in [35], and naive Agreement [54, 61, 39]. All of these methods, like ALine, are based on the softmax outputs of the model. See Appendix A for more details about previous methods.

We implement the version of ATC that performed best in the paper, i.e. with negative entropy as the score function and temperature scaling to calibrate the models in-distribution. Although DOC was deemed the best method in Guillory et al. [31], we use DOC-Feat since DOC requires information from multiple OOD datasets. For ATC, DOC, and AC, consistent with the experimental design in Garg et al. [28], we report the best number achieved between before versus after temperature scaling. We also compare with the more recent, ProjNorm by Yu et al. [87] which showed stronger linear correlation with OOD accuracy than Rotation [23] and ATC [29]. We compare with this method separately in Section 5.1, as they do not provide a way to directly estimate the OOD accuracy.

| Dataset | ALine-D* | ALine-S* | ATC | AC | DOC | Agreement |
|---|---|---|---|---|---|---|
| CIFAR-10.1 | **1.11** | 1.17 | 1.21 | 4.51 | 3.87 | 5.98 |
| CIFAR-10.2 | **3.93** | **3.93** | 4.35 | 8.23 | 7.64 | 5.42 |
| ImageNetV2 | 2.06 | 2.08 | **1.14** | 66.2 | 11.50 | 6.70 |
| CIFAR-10C-Fog | **1.45** | 1.75 | 1.78 | 4.47 | 3.93 | 3.47 |
| CIFAR-10C-Snow | 1.32 | 1.97 | **1.31** | 5.94 | 5.49 | 2.57 |
| CIFAR-10C-Saturate | **0.41** | 0.77 | 0.69 | 2.03 | 1.51 | 4.14 |
| fMoW-WILDS | **1.30** | 1.44 | 1.53 | 2.89 | 2.60 | 8.99 |
| RxRx1-WILDS | **0.27** | 0.52 | 2.97 | 2.46 | 0.75 | 8.69 |
| Camelyon17-WILDS | **5.47** | 8.31 | 11.93 | 13.30 | 13.57 | 6.79 |
| iWildCam-WILDS | 4.95 | 6.01 | 12.12 | **4.46** | 5.02 | 7.35 |

**Table 2:** Mean Absolute Error (MAE) of the OOD accuracy predictions with % as units. ALine-D outperforms other methods on both shifts where we do and do not see accuracy-on-the-line. * denotes our methods.

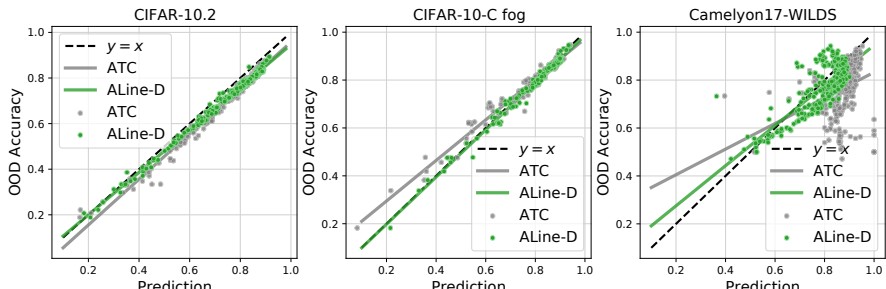

**Figure 3:** Prediction vs. OOD accuracy. We observe the scatter plot of prediction vs. OOD accuracy of ALine-D and ATC, the second best performing method from Table 2. We observe that our linear fit is closer to the diagonal, as ATC underperforms on models that have low OOD accuracy.

## 5.1 Main results: comparison to other methods.

In Table 2, we observe that ALine-D generally outperforms other methods on datasets where agreement-on-the-line holds. On ImageNet to ImageNetV2 and CIFAR-10 to CIFAR-10C-Snow, ATC performs marginally better. As can be seen in Figure 3, ATC generally cannot accurately predict the model's OOD performance for models that do not perform very well. This is consistent with experimental results in [29] and [87]. On the other hand, ALine perform equally well on "bad" models land "good" models. In some sense, given a collection of models where we are interested in the performance of each, ATC, AC, DOC-Feat, and Agreement only utilize information from the model of interest, whereas ALine utilizes the collective information from all models for each individual prediction.

As expected, on datasets where we do not observe a linear correlation between ID and OOD agreement (and accuracy), ALine does not perform very well, with a mean absolute estimation error of around 5%. Interestingly, the other methods also do not perform very well on these datasets. No method successfully predicts the OOD accuracy for every distribution shift. The advantage of ALine is that there is a concrete way to verify when the method will successfully predict the OOD accuracy (i.e. check whether agreement is on the line). Other prediction methods do not have any way of characterizing when they will be successful. Finally, we note that ALine-D actually surpasses previous methods even when accuracy-on-the-line does not hold, suggesting that the algorithm has additional beneficial properties that require further study.

## 5.2 Correlation analysis

Rather than predicting OOD accuracy, it could be useful to have a metric that just strongly correlates with the OOD accuracy, if the application simply requires an understanding of relative performance such as model selection. Recently, Yu et al. [87] proposed ProjNorm, a measurement they show has a very strong linear correlation with OOD accuracy, moreso than other recent methods including Rotation [23] and ATC [29]. To compare Aline-D with ProjNorm, we replicate the CIFAR-10C study in Yu et al. [87],

| Dataset | ALine-D | | ProjNorm | |
|---|---|---|---|---|
| | $\rho$ | $R^2$ | $\rho$ | $R^2$ |
| CIFAR-10C | **0.995** | **0.974** | 0.98 | 0.973 |

**Table 3:** Correlation analysis. We compare the coefficients of determination ($R^2$) and rank correlations ($\rho$) between ALine-D and ProjNorm, a metric shown to have stronger correlation than ATC and Rotation.

where they train a ResNet18 model and predict its performance across all corruptions and severity levels of CIFAR-10C (See their Table 1 in [87]). Since ALine-D is an algorithm that requires a set of models for prediction, we use the 29 pretrained models from the CIFAR-10 testbed of Miller et al. [59], as the other models in the set. We look at the linear correlation of the estimates of OOD accuracy and the true accuracy across all corruptions and find that ALine-D achieves *stronger* correlation than ProjNorm (Table 3). See Appendix E for more experimental details.

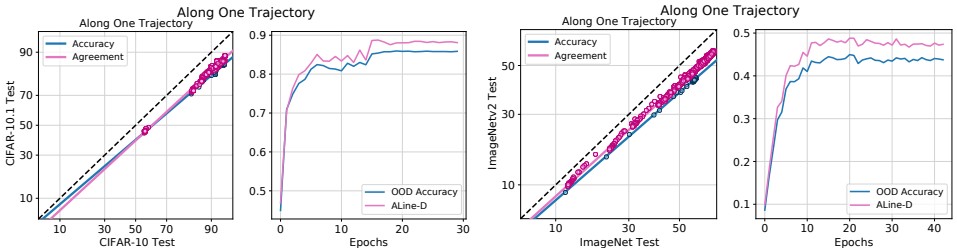

**Figure 4:** ALine-D tracks OOD accuracy across training epochs with a MAE of 2.19% for CIFAR10.1 and 3.61% for ImageNetv2.

## 5.3 Estimating performance along a training trajectory

We assess whether ALine can be utilized even in situations where the practioner only cares about the performance of a few models. In such situations, one could efficiently gather many models by training a single model and saving checkpoints along the way. We analyze whether our phenomenon is helpful for predicting such highly correlated hypotheses, instead of independently trained models. In Figure 4, we collect the ID and OOD test predictions every 5 epochs across CIFAR-10 training of a single ResNet18 model and every epoch across ImageNet training of a ResNet50 model. We see that even the agreement between every pair of checkpoints of a model across training is enough to get a good linear fit that matches the slope and bias of ID vs. OOD accuracy. By applying ALine-D to these checkpoints, we get a very good estimate of the OOD performance of the model across training epochs though slightly worse than for a collection of independently trained models. This suggests that given a model of interest, ALine does not require practitioners to train a large number of models, but just train one and save its predictions across training iterations. We do a more careful ablation study in Appendix G, looking at the number of models required for close estimates of accuracy.

## 6 Conclusion

The contributions of this work are two-fold. First, we observe the agreement-on-the-line phenomenon, and show that it correlates strongly with accuracy-on-the-line over a range of datasets and models. We also highlight that certain aspects of this phenomenon, namely the fact that the slope and bias of the linear fit is largely the *same* across agreement and accuracy, are specific to neural networks, and thus fundamentally seem connected to these classes of models. Second, using this empirical phenomenon, we propose a surprisingly simple but effective method for predicting OOD accuracy of classifiers, while only having access to unlabeled data from the new domain (and one that can be "sanity checked" via testing whether agreement-on-the line holds). Our method outperforms existing state-of-the-art approaches to this problem. Importantly, we do *not* claim that this phenomenon is universal, but we found it to be true across an extensive range of neural networks and OOD benchmarks that we experimented on. In addition to its practical relevance, this observation itself reveals something very interesting about the way neural networks learn, which we leave for future study.

## Acknowledgments and Disclosure of Funding

We thank Rohan Taori and Saurabh Garg for valuable discussions regarding the model testbeds and temperature scaling used in this work, respectively. Christina Baek was supported by a Presidential Fellowship sponsored by Carnegie Mellon University. Yiding Jiang was supported by funding from the Bosch Center for Artificial Intelligence. Aditi Raghunathan was supported by an Open Philanthropy AI Fellowship.

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
