# A   Baseline Methods

In Section 5, we compare ALine-S/D to ATC, DOC-Feat, AC, and Agreement. We detail these methods below. Given model $h$ (the output after softmax) and unlabeled OOD data, the algorithms are

**Average Threshold Confidence (ATC)**

$$\text{ATC}(h) = \frac{1}{|X_{OOD}|} \sum_{x \in X_{OOD}} \mathbf{1}\{s(h(x)) > t\} \tag{7}$$

where $s(h(x)) = \sum_j h_j(x) \log(h_j(x))$ and $t$ satisfies

$$\frac{1}{|X_{val}|} \sum_{x \in X_{val}} \mathbf{1}\{s(h(x)) < t\} = 1 - \widehat{\text{Acc}}_{\text{ID}}(h) \tag{8}$$

**DOC-Feat**

$$\text{DOC}(h) = \widehat{\text{Acc}}_{\text{ID}}(\mathsf{h}) + \frac{1}{|X_{Val}|} \sum_{x \in X_{Val}} \max_k h_k(x) - \frac{1}{|X_{OOD}|} \sum_{x \in X_{OOD}} \max_k h_k(x) \tag{9}$$

**Average Confidence (AC)**   Given model $h$ and unlabeled OOD data,

$$\text{AC}(h_\theta) = \frac{1}{|X_{OOD}|} \sum_{x \in X_{OOD}} \max_k h_k(x) \tag{10}$$

**Agreement**   Given a pair of models $h, h'$, Agreement simply predicts their average OOD accuracy to be the agreement between the models $\widehat{\text{Agr}}_{\text{OOD}}(h, h')$.

# B   Theoretical Analysis from Miller et al.

To get a better understanding of the agreement-on-the-line phenomenon, we replicate a theoretical experiments in Miller et al. [59] using the same set of neural networks from their testbed. Specifically, we look at CIFAR-10 with different added Gaussian noise (their Figure 4).

## B.1   Matching gaussian noise

Miller et al. [59] conduct a theoretical analysis on a toy gaussian mixture model to better understand the accuracy-on-the-line phenomena. From their analysis, they predict that accuracy-on-the-line occurs if the covariances of the ID and OOD data are the same up to some constant scaling factor. Inspired by this, they show that accuracy-on-the-line holds stronger on CIFAR-10 data corrupted with gaussian noise that matches the covariance of CIFAR-10 test data versus isotropic gaussian noise. Interestingly, even for this simple setting, we similarly observe that the ID vs OOD agreement trend is stronger on covariance matched gaussian noise.

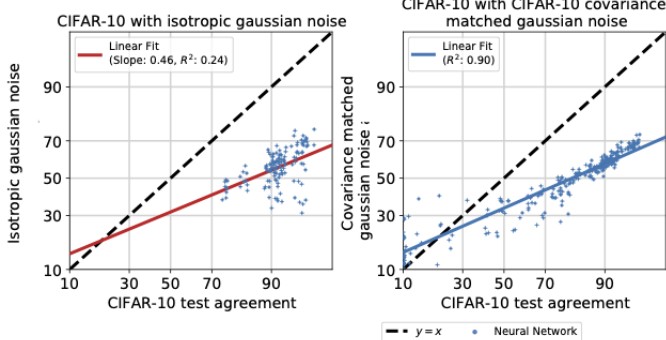

**Figure 5:** We look at ID vs OOD agreement of CIFAR-10 with isotropic gaussian noise versus covariance matched gaussian noise. See Figure 4 in Miller et al. [59]

# C    Correlation results on more datasets

To verify the agreement-on-the-line phenomenon, we compare the linear correlation between ID vs OOD accuracy and agreement across a variety of distribution shifts. Given a set of $n$ models, we plot the ID vs OOD accuracy of each model in the set. For agreement, we randomly pair each model with another model in the set, and plot the ID vs OOD agreement of these $n$ pairs. We provide $R^2$ values in the figure legends.

**Datasets**    In addition to the 8 datasets from the main body, we also observe the trend on other CIFAR10C corruptions [34]. For each corruption, we evaluate the models over data from all 5 severity levels (both in Figure 1 of main body and appendix).

We also look at shifting from CIFAR-10 to CINIC-10 [19] and CIFAR-10 to STL-10 [16] which are shifts from changes in the image source. Specifically, the CINIC-10 test dataset contains both CIFAR-10 Test data and a selection of ImageNet images for CIFAR-10 class labels downsampled to $32 \times 32$. We only consider the downsampled ImageNet data as the OOD dataset. Similarly, STL-10 contains processed ImageNet images for CIFAR-10 class labels. Since STL-10 is an unsupervised learning dataset, we only utilize the labeled subset of STL-10 as the OOD dataset. Additionally, STL-10 only contains 9 of the 10 CIFAR-10 classes, so we restrict the dataset to just those 9 classes.

Finally, we add results for a real-world shift from batch effects in images of cells in RxRx1-WILDS and a reading comprehension dataset Amazon-SQuAD [58] which looks at the reading comprehension performance of models trained on paragraphs derived from Wikipedia articles on Amazon product reviews.

**Models**    For ImageNetV2, we evaluate 49 ImageNet pretrained models from the timm [82] package. See their repository for more details about the models. For RxRx1-WILDS we trained 36 models of varying architecture and hyperparameters. Specifically, we vary weight decay between $[10^{-1}, 10^{-2}, 10^{-3}, 10^{-4}, 10^{-5}]$ and optimizers between SGD, Adam, and AdamW. See D for architecture details.

Finally, we utilize all independently trained models from the CIFAR10, iWildCam-WILDS, fMoW-WILDS, and Camelyon17-WILDS testbeds created and utilized by [59]. The hyperparameters used to train these models are explained in great detail in Appendix B.2 of Miller et al. [59]. The architectures of all models evaluated for the experiment are described in more detail in Appendix D.

**Pretrained vs Not Pretrained**    Miller et al. [59] showed that for some shifts, the ID vs OOD accuracy of ImageNet pretrained models follow a different linear trend than models trained from scratch in-distribution. In Figure 1 of the main body, we do not distinguish between pretrained and from scratch models as the trends for pretrained and from scratch models were the same for the 8 datasets we chose. Below, we divide results between pretrained and not pretrained models to be more precise. [3] [4]

---

[3] The fMoW-WILDS testbed also contains two models pretrained on CLIP

[4] For CINIC-10 and STL-10, we only look at from scratch models as the shifted images are derived from ImageNet.

## C.1 Over randomly initialized models

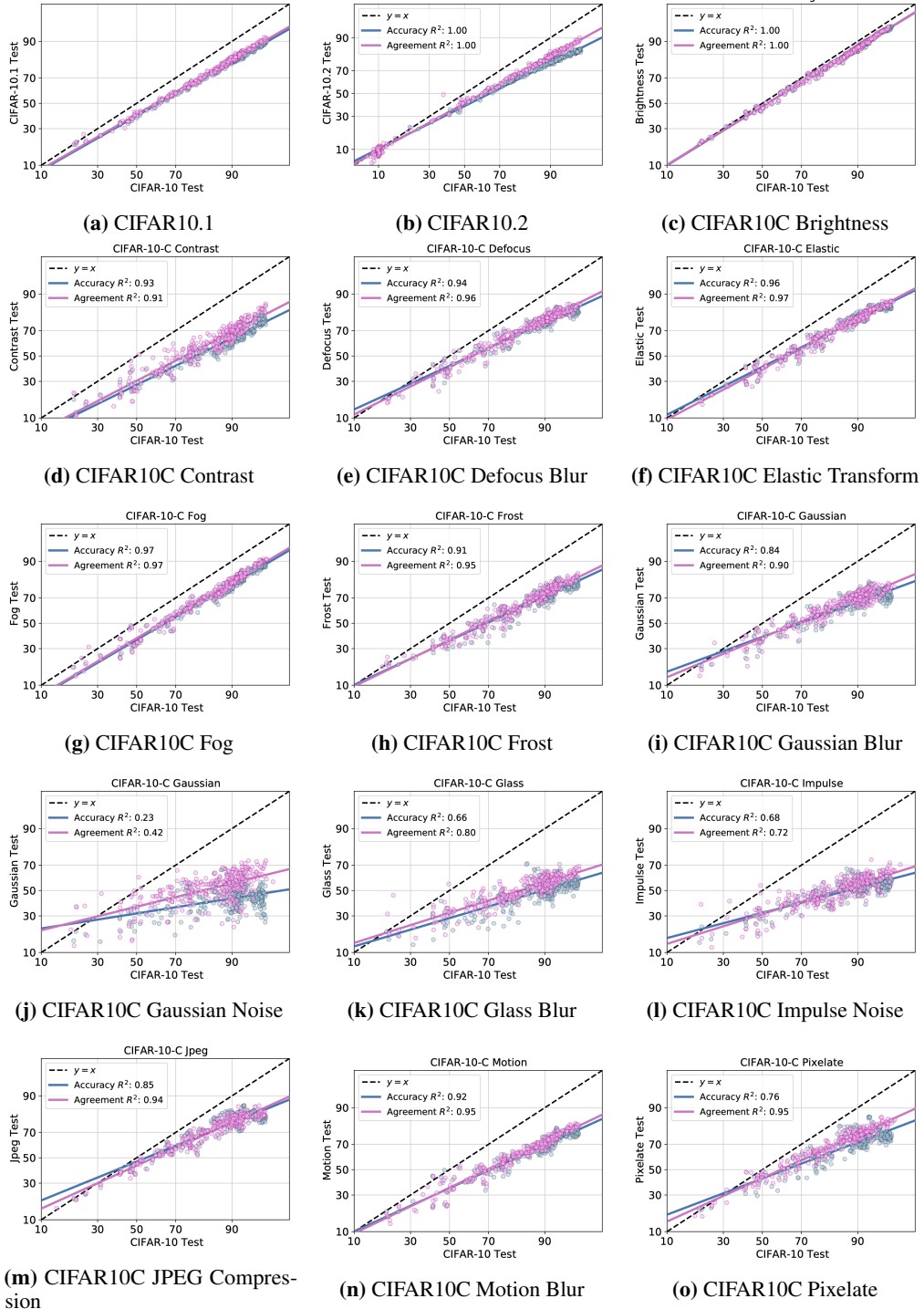

**(a)** CIFAR10.1

**(b)** CIFAR10.2

**(c)** CIFAR10C Brightness

**(d)** CIFAR10C Contrast

**(e)** CIFAR10C Defocus Blur

**(f)** CIFAR10C Elastic Transform

**(g)** CIFAR10C Fog

**(h)** CIFAR10C Frost

**(i)** CIFAR10C Gaussian Blur

**(j)** CIFAR10C Gaussian Noise

**(k)** CIFAR10C Glass Blur

**(l)** CIFAR10C Impulse Noise

**(m)** CIFAR10C JPEG Compression

**(n)** CIFAR10C Motion Blur

**(o)** CIFAR10C Pixelate

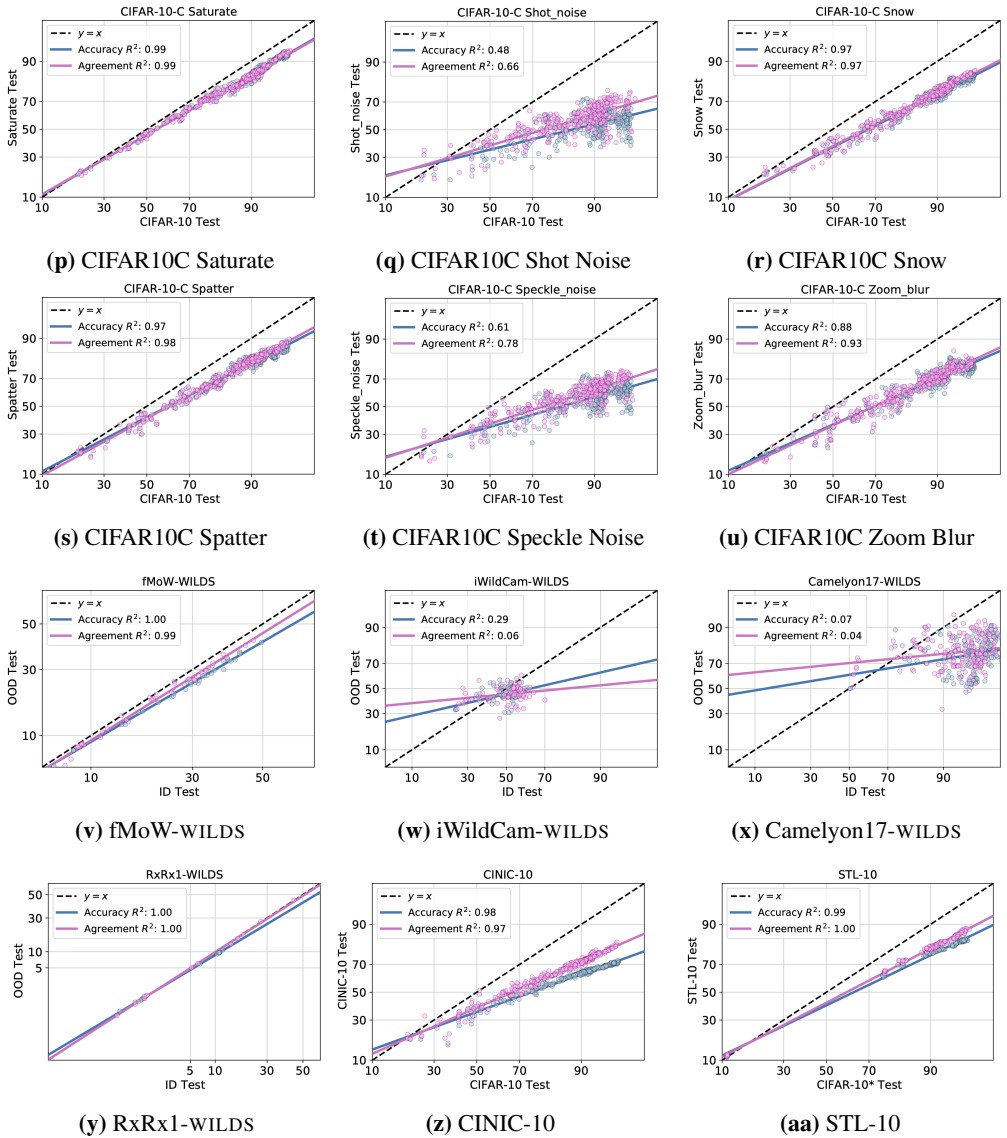

**(p)** CIFAR10C Saturate      **(q)** CIFAR10C Shot Noise      **(r)** CIFAR10C Snow

**(s)** CIFAR10C Spatter      **(t)** CIFAR10C Speckle Noise      **(u)** CIFAR10C Zoom Blur

**(v)** fMoW-WILDS      **(w)** iWildCam-WILDS      **(x)** Camelyon17-WILDS

**(y)** RxRx1-WILDS      **(z)** CINIC-10      **(aa)** STL-10

## C.2 Over models pretrained on ImageNet

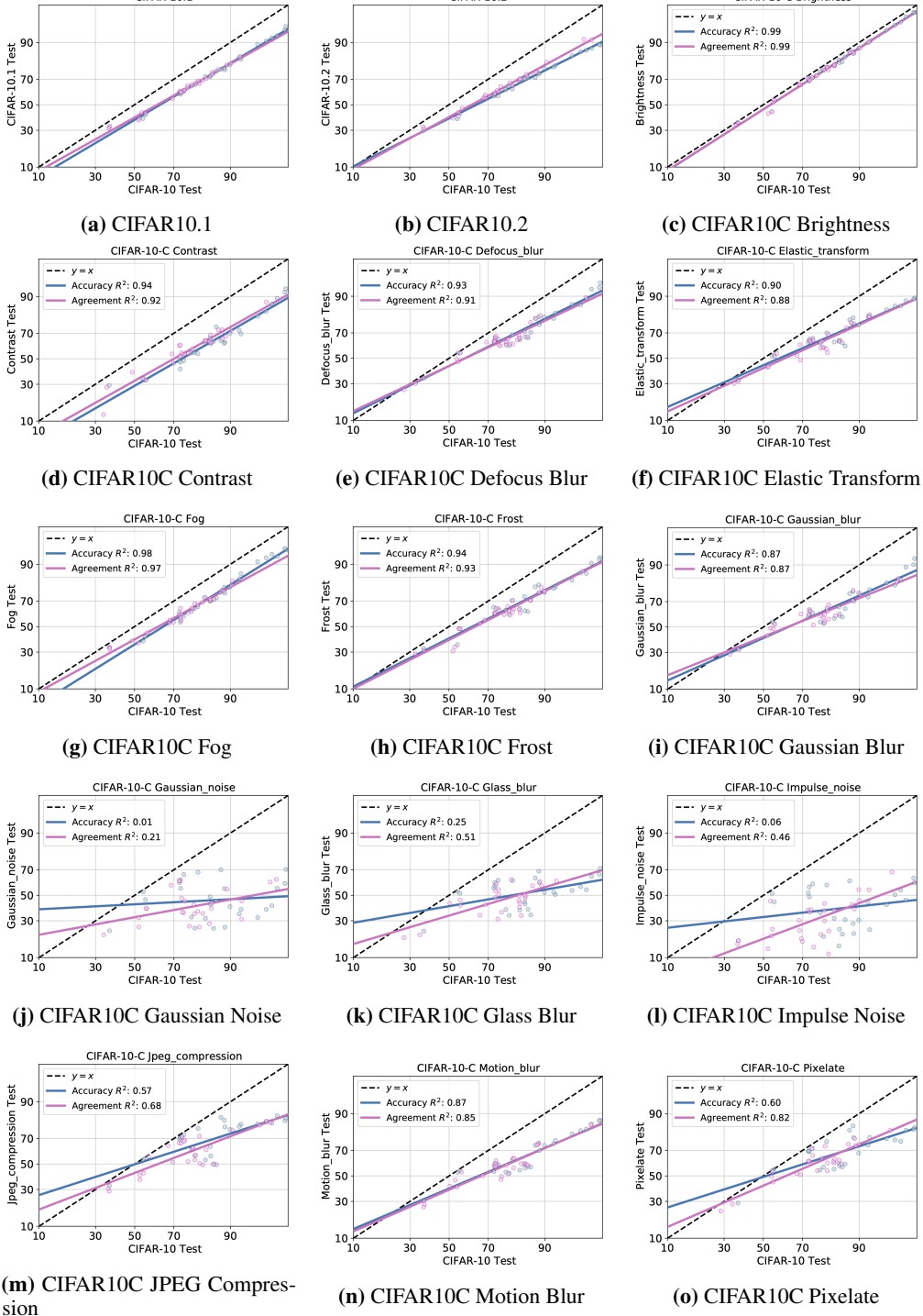

(a) CIFAR10.1     (b) CIFAR10.2     (c) CIFAR10C Brightness

(d) CIFAR10C Contrast     (e) CIFAR10C Defocus Blur     (f) CIFAR10C Elastic Transform

(g) CIFAR10C Fog     (h) CIFAR10C Frost     (i) CIFAR10C Gaussian Blur

(j) CIFAR10C Gaussian Noise     (k) CIFAR10C Glass Blur     (l) CIFAR10C Impulse Noise

(m) CIFAR10C JPEG Compression     (n) CIFAR10C Motion Blur     (o) CIFAR10C Pixelate

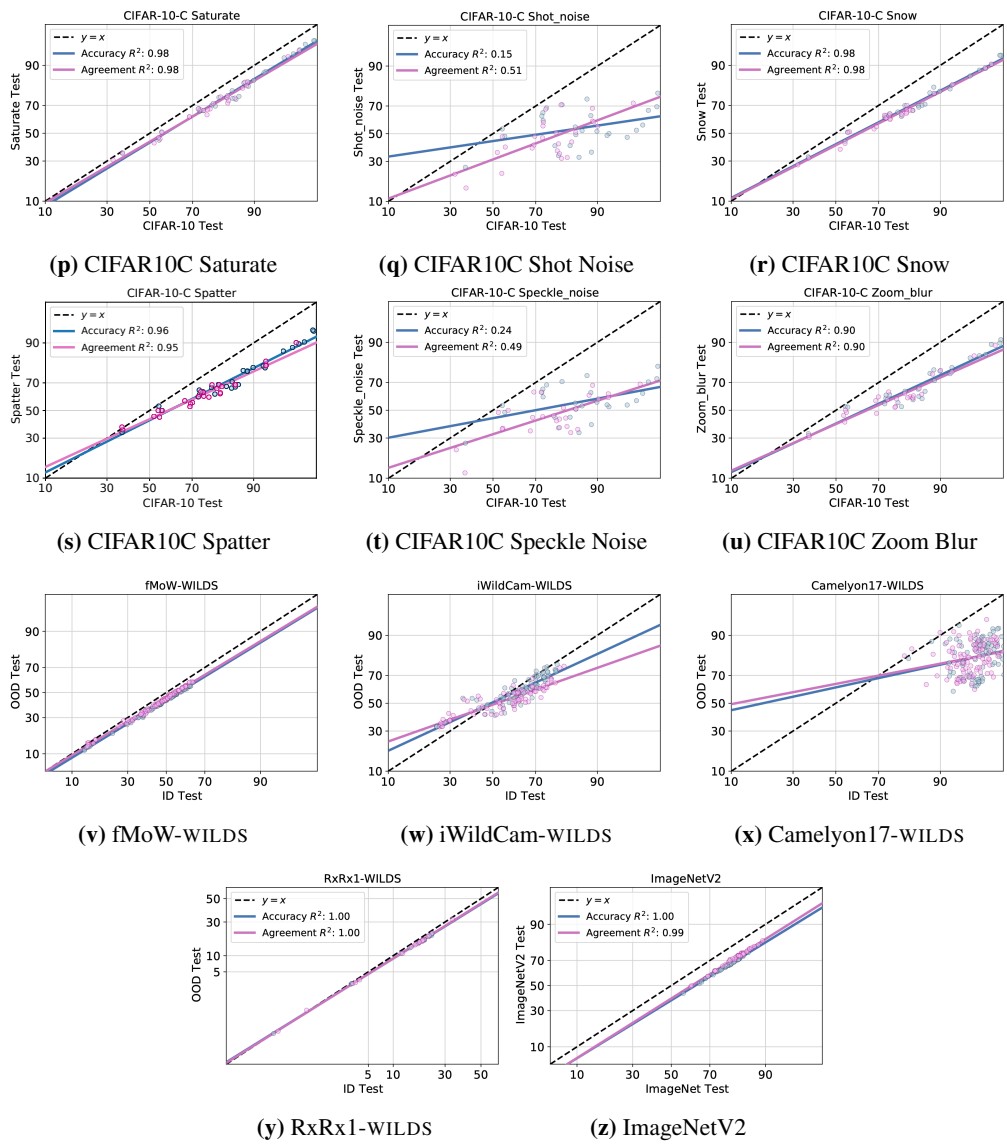

**(p)** CIFAR10C Saturate     **(q)** CIFAR10C Shot Noise     **(r)** CIFAR10C Snow

**(s)** CIFAR10C Spatter     **(t)** CIFAR10C Speckle Noise     **(u)** CIFAR10C Zoom Blur

**(v)** fMoW-WILDS     **(w)** iWildCam-WILDS     **(x)** Camelyon17-WILDS

**(y)** RxRx1-WILDS     **(z)** ImageNetV2

## C.3 NLP Datasets

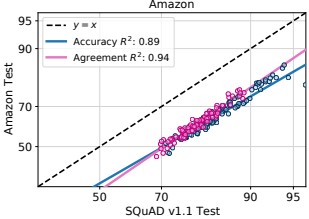

**Figure 8:** Amazon Q and A

Here, we plot the ID vs OOD F1 score of the models versus the ID (SQuAD) vs OOD (Amazon) F1-agreement, where F1-agreement is simply the average F1 score where one of the models is treated

as the labels or the ground truth. Specifically,

$$\text{Agr}_{F1}(h, h') = \frac{F1(h, h') + F2(h', h)}{2} \tag{11}$$

This is analogous to how agreement relates to accuracy i.e. agreement can be thought of as accuracy where one of the models is treated as the ground truth. We look at 99 models from CodaLab. Note that the linear correlation of agreement is strong with $R^2$ is $0.94$. ALine-D achieves a mean estimation error of $0.028$.

## C.4 Only Neural Networks

Here we observe the ID vs OOD accuracy and agreement trend of model families other than neural networks. In the main body of the paper, we illustrated how the agreement-on-the-line phenomenon is specific to neural networks on CIFAR10-Fog, a synthetic shift. Below, we illustrate this further on a data replication shift, CIFAR10.2, and a real-world shift, fMoW-WILDS. Note that the slope of the ID vs OOD agreement trend of neural network models is closest to the slope of the ID vs OOD accuracy trend. Excluding neural networks, we observe that the agreement trend of random feature models [15] also has a similar slope to that of the accuracy trend for select shifts such as CIFAR-10.2.

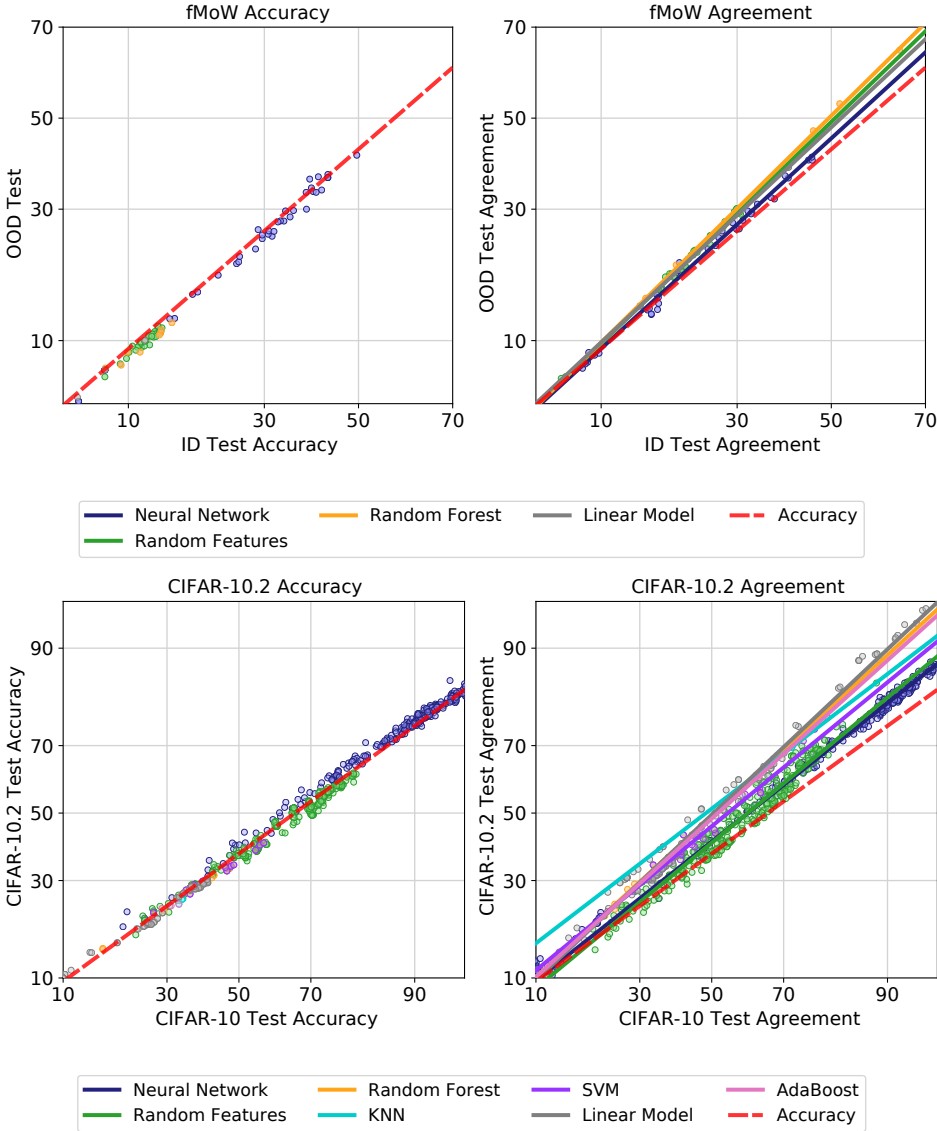

**Figure 9:** ID vs OOD Accuracy/Agreement of different model families. Given $n$ models, we plot the ID vs OOD accuracy of each model. For agreement, we plot the agreement of $n$ random pairs between models of the same family. The linear trend of ID vs OOD accuracy (in red dashes) is computed using models from all model families.

# D Model Architectures

We list the types of architectures and the corresponding number of models for each testbed.

**Table 4:** CIFAR-10 Testbed [59]: 467 models total, out of which 29 are pretrained on ImageNet.

| Architecture | Number of models |
| --- | --- |
| DenseNet121 [36] | 21 |
| DenseNet169 [36] | 8 |
| EfficientNetB0 [77] | 13 |
| ResNet18 [33] | 13 |
| ResNet50 [33] | 18 |
| ResNet101 [33] | 7 |
| PreActResNet18 [32] | 63 |
| PreActResNet34 [32] | 9 |
| PreActResNet50 [32] | 11 |
| PreActResNet101 [32] | 4 |
| ResNeXT $2 \times 64$d [83] | 12 |
| ResNeXT $32 \times 4$d [83] | 8 |
| ResNeXT $4 \times 64$d [83] | 1 |
| RegNet X200 [66] | 11 |
| RegNet X400 [66] | 13 |
| RegNet Y400 [66] | 5 |
| VGG11 [72] | 16 |
| VGG13 [72] | 13 |
| VGG16 [72] | 13 |
| VGG19 [72] | 12 |
| ShuffleNetV2 [53] | 56 |
| ShuffleNetG2 [88] | 13 |
| ShuffleNetG3 [88] | 8 |
| AlexNet [44] | 2 |
| MobileNet [70] | 12 |
| MobileNetV2 [70] | 13 |
| PNASNet-A [48] | 13 |
| PNASNet-B [48] | 13 |
| PNASNet-5-Large [48] | 3 |
| SqueezeNet [37] | 3 |
| SENet18 [40] | 13 |
| GoogLeNet [75] | 20 |
| DPN26 [11] | 8 |
| DPN92 [11] | 2 |
| Myrtlenet [Repo] | 1 |
| Xception [12] | 3 |

**Table 5:** ImageNet Testbed: 49 models total from the timm package [82] and Torchvision [link].

| Architecture | Number of models |
| --- | --- |
| Adversarial Inception v3 [45] | 1 |
| AlexNet [44] | 1 |
| BEiT [2] | 1 |
| BoTNet [73] | 1 |
| CaiT [80] | 1 |
| CoaT [84] | 2 |
| ConViT [20] | 3 |
| ConvNeXT [50] | 1 |
| CrossViT [8] | 9 |
| DenseNet [36] | 3 |
| DLA [86] | 10 |
| EfficientNet [77] | 1 |
| HaloNet [81] | 1 |
| NFNet [7] | 1 |
| ResNet [33] | 10 |
| ResNeXT [83] | 1 |
| Inception v3 [76] | 1 |
| VGG [72] | 1 |

**Table 6:** FMoW Testbed [59]: 161 models total, out of which 37 are pretrained on ImageNet and 2 are CLIP pretrained models.

| Architecture | Number of models |
| --- | --- |
| ResNet [33] | 40 |
| ResNeXT [83] | 18 |
| AlexNet [44] | 11 |
| DPN68 [11] | 15 |
| DenseNet121 [36] | 11 |
| GoogLeNet [75] | 8 |
| Xception [12] | 11 |
| ShuffleNet [88] | 10 |
| MobileNetV2 [70] | 8 |
| VGG [72] | 15 |
| PNASNet [48] | 2 |
| Squeezenet [37] | 11 |
| ViT [25] | 1 |

**Table 7:** iWildCam-WILDS Testbed: 157 models total, out of which 81 are pretrained.

| Architecture | Number of models |
|---|---|
| AlexNet [44] | 30 |
| ShuffleNetV2 [53] | 30 |
| ResNeXT [83] | 5 |
| ResNet [33] | 38 |
| VGG [72] | 5 |
| SqueezeNet [37] | 5 |
| MobileNetV2 [70] | 30 |
| PNASNet [48] | 4 |
| Xception [12] | 5 |
| DenseNet [36] | 5 |

**Table 8:** Camelyon17-WILDS Testbed: 269 models total out of which 100 are pretrained on ImageNet.

| Architecture | Number of models |
|---|---|
| ResNet [33] | 29 |
| SqueezeNet [37] | 27 |
| ShuffleNetV2 [53] | 29 |
| VGG [72] | 28 |
| AlexNet [44] | 29 |
| MobileNetV2 [70] | 29 |
| ResNeXT [83] | 27 |
| DenseNet [36] | 28 |
| Xception [12] | 28 |
| PNASNet [48] | 15 |

**Table 9:** RxRx1-WILDS Testbed: 36 models total out of which 16 are pretrained on ImageNet.

| Architecture | Number of models |
|---|---|
| ResNet18 [33] | 9 |
| ResNet50 [33] | 21 |
| DenseNet121 [36] | 6 |

# E    Section 5: Experimental Details

## E.1    Details for Experiment 5.2: Correlation analysis

We replicate the correlation analysis experiment in Table 1 of Yu et al. [87] to compare the prediction performance of ALine-D versus ProjNorm. Essentially, we want to see how strong the linear correlation is between the estimate of OOD accuracy versus the true OOD accuracy by looking at the coefficients of determination $R^2$ and rank correlations $\rho$ of the fit. We use the GitHub repository of ProjNorm [87] found at `https://github.com/yaodongyu/ProjNorm` to replicate their correlation analysis experiment found in their Table 1. Using their repository, we train a base ResNet18 model for 20 epochs with their default hyperparameters

- Batch Size: 128
- Learning Rate: 0.001
- Weight Decay: 0
- Optimizer: SGD with Momentum 0.9
- Pretrained: True

using cosine learning rate decay [51]. The repository uses the default implementation of ResNet18 by torchvision. The CIFAR-10 images are resized to be $224 \times 224$, then normalized. To compute ProjNorm, we use the repository to train a reference ResNet18 model using the pseudolabels of the base model for 500 iterations with the same hyperparameters as the base model. We use the ALine-D algorithm to predict the OOD accuracy of the base model. ALine-D requires a model set $\mathcal{H}$ with at least 3 models so that the linear system of equations (Line 9 in Algorithm 1, Equation 6 in main body) has a unique solution. We use the 29 pretrained models from the CIFAR10 testbed as the other models in the model set.

## E.2    Details for Experiment 5.3: Performance along a training trajectory

We train a ResNet18 model from scratch (no pretrained weights) on CIFAR10 for 150 epochs using the following hyperparameters:

- Batch Size: 100
- Learning Rate: 0.1
- Weight Decay: $10^{-5}$
- Optimizer: SGD with Momentum 0.9
- Pretrained: False

We decay the learning rate by 0.1 at epoch 75 and epoch 113. We use simple data augmentation consisting of RandomCrop with padding$= 4$, RandomHorizontalFlip, and Normalization. We modify ResNet18 to take in $32 \times 32$ input images.

During training, we save the model weights every 5 training epochs. Given this collection of models, we estimate the CIFAR-10.1 accuracy of each one of these models using the ALine-D procedure.

## E.3    Hardware

All experiments were conducted using GeForce GTX 1080 and 2080 graphics cards.

## F    The relationship between agreement and accuracy

This work is related to Jiang et al. [39], which shows that the agreement between two models of the *same architecture* trained with *different random seeds* is approximately equal to the average ID test accuracy of models if the ensemble consisting of the models is *well-calibrated*. They call this equality between accuracy and agreement Generalization Disagreement Equality (GDE).

Let us ignore the probit transform for a moment and assume the linear correlation between ID vs OOD accuracy and agreement are strong without it. In this simplified scenario, agreement-on-the-line implies that, for datasets where both agreement and accuracy are strongly linearly correlated, if ID agreement of a pair of models is equal to their average ID accuracy, then their OOD agreement is equal to their OOD accuracy. Formally, when a shift satisfies accuracy-on-the-line, we know by agreement-on-the-line that for any two models trained on ID samples $h, h' \in \mathcal{H}$, the following equations are approximately satisfied (ignoring probit transform)

$$\frac{\mathsf{Acc_{OOD}}(h) + \mathsf{Acc_{OOD}}(h')}{2} = a \cdot \frac{\mathsf{Acc_{ID}}(h) + \mathsf{Acc_{ID}}(h')}{2} + b \tag{12}$$

$$\text{and} \quad \mathsf{Agr_{OOD}}(h, h') = a \cdot \mathsf{Agr_{ID}}(h, h') + b \implies \tag{13}$$

$$\underbrace{\frac{\mathsf{Acc_{OOD}}(h) + \mathsf{Acc_{OOD}}(h')}{2} - \mathsf{Agr_{OOD}}(h, h')}_{\text{OOD Gap}} = a \cdot \underbrace{\left( \frac{\mathsf{Acc_{ID}}(h) + \mathsf{Acc_{ID}}(h')}{2} - \mathsf{Agr_{ID}}(h, h') \right)}_{\text{ID Gap}} \tag{14}$$

for some slope $a$ and intercept $b$. Thus, if the "ID gap" between accuracy and agreement is 0, then the "OOD gap" is also 0. This may suggest something about calibration on shifts where accuracy-on-the-line holds: if the ensemble of a pair of models is well calibrated ID, then by agreement-on-the-line GDE also holds OOD.

However, agreement-on-the-line goes beyond these results in two ways: (i) agreement between models with *different architectures* and (ii) agreement between different checkpoints on the *same training run* is also on the ID vs OOD agreement line. Jiang et al. [39] does not guarantee GDE holds for these cases. As can be seen in Figure 10, for most pairs of models, the ID and OOD gaps between probit scaled accuracy and agreement are not equal to 0 i.e. GDE does not occur ID or OOD. Indeed, understanding why agreement-on-the-line holds requires going beyond the theoretical conditions presented in the prior work [39] which do not hold for this expanded set of models.

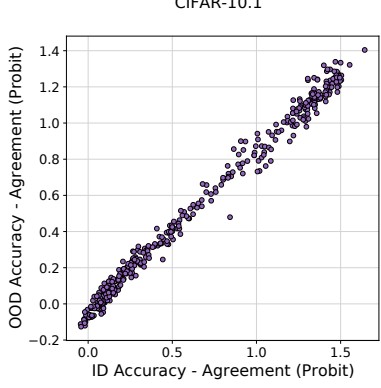

**Figure 10:** We plot the ID (CIFAR-10 Test) vs OOD (CIFAR 10.1) Test gap for 468 pairs of models randomly sampled from the CIFAR10 testbed. We observe that due to agreement-on-the-line, we observe a strict linear correlation (Eq. 9). However, not all pairs satisfy GDE (ID or OOD gap is not close to 0).

*Is agreement-on-the-line a result of neural network performing well?* In Figures 11 we look at the trend between ID agreement vs accuracy to observe whether the accuracy of the models in the collection plays a role in the success of ALine-D. Note that naturally, the gap between agreement and average accuracy is smaller for pairs of models that perform well. However, this does not necessarily correspond with better ALine-D performance on such models. On the right figure of 11, we plot the right hand side of ALine-D's Equation 6 versus the left hand side ("adjusted agreement"), and observe that all pairs lie close to the diagonal, regardless of how large the gap $\mathsf{Acc}_{OOD} - \mathsf{Agr}_{OOD}$ is before

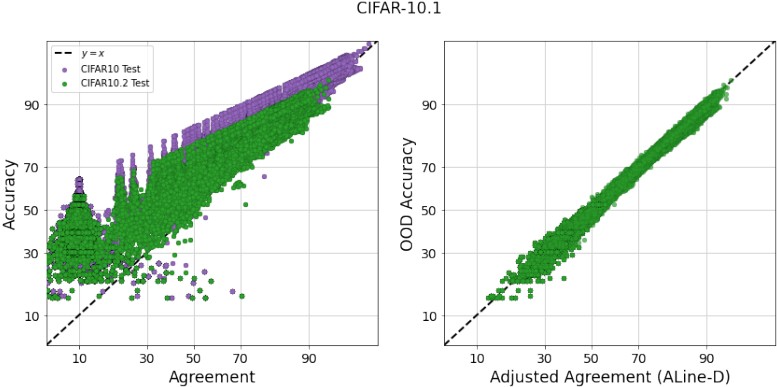

**Figure 11:** We plot the agreement vs average accuracy (after probit scaling) for every pair of 467 models trained on CIFAR10. We observe that accuracy is not exactly equal to agreement, but the agreement of better performing pairs are closer to its accuracy. However, after the adjustment provided in the left hand side of Equation 6, the agreement matches OOD accuracy.

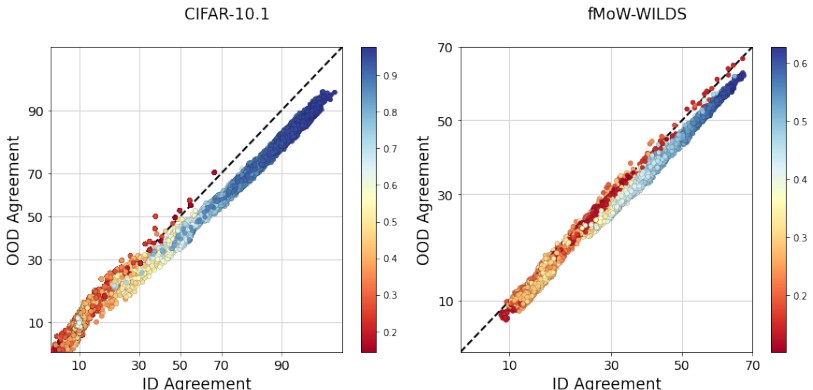

**Figure 12:** We plot the ID vs OOD agreement of each pair of models in the corresponding testbeds. The color of the dot represents the average ID test accuracy of the pair of models.

the adjustment. This is to say that, all pairs of varying performance have strong linear correlation between ID vs OOD agreement, which we show in Figure 12.

# G    Ablation Study

## G.1    How many models are necessary for ALine-D to make accurate predictions?

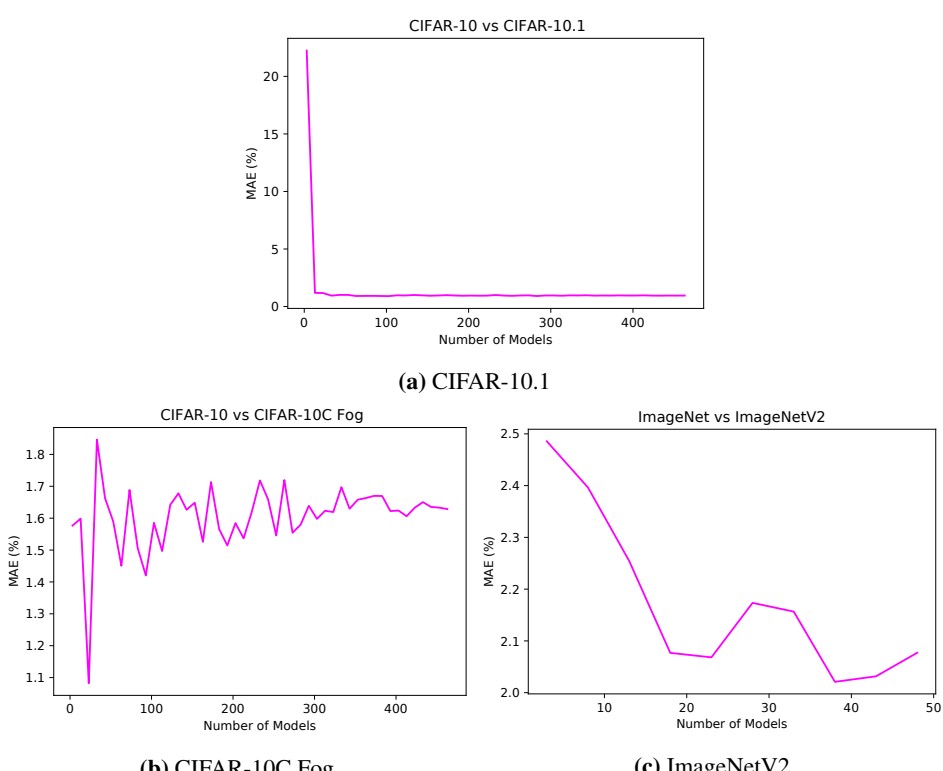

**(a)** CIFAR-10.1

**(b)** CIFAR-10C Fog

**(c)** ImageNetV2

**Figure 13:** Performance of ALine-D over model sets of different sizes. We observe the MAE (in %) of the ALine-D estimates of the OOD accuracy of each model in the model set. On the x axis, we vary the size of the model set. We look at CIFAR-10.1, CIFAR-10C Fog, and ImageNetV2 in particular. For each plot, we average the MAE over 3 seeds.

ALine-D is an algorithm that requires at least 3 models in the model set so that the system of linear equations (Line 9 in Algorithm 1) has a unique solution. Additionally, it may generally require more models for the slope and intercept of the agreement trend to match the slope and intercept of the accuracy trend. We observe the MAE (in %) of the ALine-D estimates of the OOD accuracy for model sets of varying sizes. For each distribution shift, we randomly sample $n$ models from the testbed to be our model set. $n$ ranges from 3 to 463 in increments of 10 for CIFAR-10 related shifts, and 3 to 48 in increments of 5 for ImageNetV2. Our ablation study shows that the success of ALine-D is not necessarily tied to the number of models. For CIFAR-10.1, we see a very quick drop in estimation error, and ALine-D performs well even for a small model set. Similarly, in ImageNetV2, we observe a decrease in estimation error as the number of models increases, however, the MAE is already pretty low from the start ($2.5\%$). On the other hand, in CIFAR-10C Fog, the estimation error does not decrease, but the error is quite low (below $1.8\%$) from the start. This short ablation study seems to indicate that ALine-D performs pretty well when agreement-on-the-line holds even for a small number of models ($< 15$ models). Additionally, it is not always the case that more models will decrease the estimation error further.

## G.2    Does a model set of varied architecture perform better than models of the same architecture?

We study whether the diversity from varying the architecture of the models in the model set improves the performance of ALine-D. We look at the performance of ALine-D on CIFAR-10.1 and CIFAR-10C Fog over two model sets sampled from the CIFAR-10 testbed: (A) 20 PreActResNet18 [32]

models, (B) 20 models of varying architecture. Similary for fMoW, we look at the performance over (A) 10 DenseNet121 [36] models , (B) 10 models of varying architecture. We randomly sample these models from the corresponding testbeds, and average our results over 10 seeds. Our results as shown below depends on the dataset. For CIFAR-10.1 and CIFAR-10C Fog, we see that ALine-D performs better on the diverse set consisting of many architecture types. On the other hand, for fMoW, ALine-D performed better on the uniform set consisting of models of one architecture.

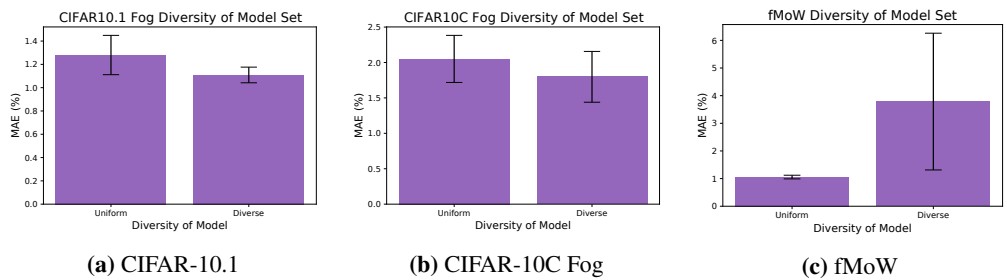

(a) CIFAR-10.1  (b) CIFAR-10C Fog  (c) fMoW

**Figure 14:** We compare the performance of ALine-D for a model set consisting of models of many architectures versus a single architecture.