# OpenReview forum: "Agreement-on-the-line: Predicting the Performance of Neural Networks under Distribution Shift"
_NeurIPS.cc/2022/Conference — NeurIPS 2022 Accept_

### Official Review · Reviewer_2n7P · 2022-07-08

**Rating:** 8
**Confidence:** 4
**Soundness:** 3 good
**Presentation:** 3 good
**Contribution:** 4 excellent

**Summary:**

The paper investigates empirically the important and practically very relevant question of OOD performance. The approach taken builds on previous work of 'accuracy-on-the-line' (which observes the occurrence of strong linear (transformed) correlation of ID vs OOD accuracy in some datasets) and adds the simultaneous evaluation of agreement between pairs of models in the OOD vs ID.
On a varied set of ID/OOD Image classification tasks, the work empirically finds that a similar linear (transformed) relationship exists or does not exists jointly for accuracy and agreement. Further the findings show a strong similarity between the slopes and bias of the agreement and accuracy ID vs OOD lines on a significant subset of the studied datasets. Interestingly outside of neural networks this similarity breaks down.
Through these observations the authors address an important practical problem of assessing existing models' performance on OOD data, critically, without the availability of labels for the OOD data.
The proposed method relies on the observed joint existence (or lack there-off) of accuracy-on-the-line and agreement-on-the-line phenomena --- such that if there is no agreement-on-the-line one can test (verify) it.
Then, by exploiting the similarity of the lines, they show how to estimate the expected OOD accuracy.

The paper proposes both a pair-model based assessment as well as a multi-model based assessment which allows for noise and potential bias reductions (e.g. from the transform mismatch).
Finally and importantly, the paper shows for a specific case that agreement-on-the-line holds also across training.




**Questions:**

see above

**Limitations:**

yes

**Strengths And Weaknesses:**

$\textbf{Strengths:}$


The paper adds simple, novel and very powerful set of observations:
* The existence (or lack there-off) of strong linear (transformed) correlation of accuracy is empirically found to go hand in hand with a strong linear (transformed) correlation of agreement between ID and OOD.
* When both exist the correlation coefficient are 'similar' (in a large subset of the cases) for neural networks examined.

These observations are interesting both theoretically (for further research) and practically --- for the pertinent question of how one may assess OOD expected accuracy of a given model given OOD, unlabeled, data.

By building on the above observations together the authors demonstrate how this practical question may be addressed:
OOD accuracy feasibility can be verified (if linear correlation is high) and carried out (assuming identical correlation coefficients between accuracy ID vs OOD line and agreement ID vs OOD line hold).

The method is straightforward and clear given the assumptions.

Throughout the paper is clearly written.

$\textbf{Weaknesses / areas of improvement:}$

1. Most important --- assessment of single model OOD performance:

In practice, it is most often the case that the practical question is for a given model --- what would be its OOD accuracy.
The authors introduce a path to address that (this is a strength of the paper) though the throughout-training agreement vs accuracy experiment, however, the authors only cursorily touch on the same-model OOD performance with a single example.
Since this a very powerful and practical setting, seeing that is generalizes across the datasets is important.
It is advisable the authors expand this experiment across the datasets and in particular for the imagent/V2 datasets.

2. Generalizability and Implications --- Is agreement-on-the-line a fundamental phenomena, does it hold outside of image classification?
It would increase the confidence in the generalizability of these finding (and their applicability) if demonstrated in the NLP context.
It would be very good to add for example a checkpointed pre-trained transformers (single model across training for simplicity) and evaluate loss and agreement loss for ID vs OOD.
Even if the phenomena does not hold in this setting that would increase the insight provided by this paper for further research.

Adding these items would change this to an even stronger paper

$\underline{Minor:}$

stylistic / typos:

line 112 --- reference 42 duplication

line 231 --- the $\Phi^{-1}$ probate transform is undefined (left for the reader to piece together)

line 233 --- summation index typo, should be $i \neq j$

line 242 --- should read 'substituting $b=...$ into (5)'

In practice (and in the appendix) the 'almost the same slope and bias' observation seems to hold to various degrees across the data examined and is left unquantified, or only quantified indirectly (e.g. through the MSE of the accuracy estimation in a subset of the cases).
consider adding a systematic quantification in the paper across the datasets (with details in the appendix) in terms of MSE OOD accuracy prediction.

---

> ### Author Response · Authors · 2022-08-02
> **Response for Reviewer 2n7P**
>
> Thank you for the detailed feedback! We’ve made several improvements to our paper based on your recommendations
>
> > Since this a very powerful and practical setting, seeing that it generalizes across the datasets is important. It is advisable for the authors expand this experiment across the datasets and in particular for the imagenet/V2 dataset.
>
> Thank you for the suggestion. We will update our submission within the discussion period with the trajectory results for ImageNetV2 for a ResNet50.  **We have updated our submission with this experiment**
>
> > Is agreement-on-the-line a fundamental phenomenon, and does it hold outside of image classification? It would increase the confidence in the generalizability of these findings (and their applicability) is demonstrated in the NLP context
>
> We added results on two additional datasets in the Appendix. First, we added a result on RxRx1, which is a cell microscopy image benchmark. We’ve also added results on large language models for Amazon, which is a reading-comprehension NLP dataset looking at Amazon product reviews  **We have updated our submission with this experiment**.
>
> > In practice (and in the appendix) the 'almost the same slope and bias' observation seems to hold to various degrees across the data examined and is left unquantified, or only quantified indirectly (e.g. through the MSE of the accuracy estimation in a subset of the cases). consider adding a systematic quantification in the paper across the datasets (with details in the appendix) in terms of MSE OOD accuracy prediction.
>
> Thank you for the feedback. We’ve provided the following experiment to test the statistical significance of the differences in slope.
> We randomly sample 1000 subsets of 10 models from the testbed and compute the corresponding difference in slope $a_{acc} - a_{agr}$. We look at the 95% confidence interval of the distribution of these slope differences for the 6 datasets where we observe a strong correlation in the main body:
>
> - CIFAR10.1 (-0.046, 0.017)
> - CIFAR10.2 (-0.120, -0.030)
> - ImageNetV2 (-0.0720, 0.061)
> - CIFAR10C-Fog(-0.077, 0.053)
> - CIFAR10C-Snow (-0.067, 0.047)
> - FMoW (-0.042, 0.030)
>
> Note that for all datasets except CIFAR10.2, the null hypothesis (difference in slopes is 0) lies within the 95% confidence interval, thus it cannot be rejected.
> We will include this experiment in the final version of the paper.
> Overall, thank you for recognizing the novelty and strengths of our work. Please let us know if there are any further questions or concerns.

---

> > ### Comment · Reviewer_2n7P · 2022-08-06
> > **Reply to authors**
> >
> > I thank the authors for their response and additions, and believe these have strengthened the paper.
> > Could you please mark in color the modifications so it would be easier to track the changes made to the paper.
> >
> > In the single trajectory (during training) there seem to be systemic deviations from the linear relationship (suspected for different phases of the stepped learning rate) --- while this does not affect the utility of the method, it would be advisable to call this out. It is unclear why coupling with different phases of training would occur, and may hold clues to the origin or limitations of the phenomena (or where one should be wary of its applicability).
> >
> > Please update when the NLP example is also added for review --- this is an exciting extension.

---

> > > ### Author Response · Authors · 2022-08-09
> > > **NLP Results + Rewording sentences**
> > >
> > > Hello,
> > >
> > > Thank you for your patience. We have updated the paper with our results on Amazon in **Appendix B.3**. Specifically, a previous work [1] also by Miller et al. showed that the ID vs OOD F1 score of question answering models *observes a strong linear trend*.  To summarize, agreement on the line phenomenon also holds for Amazon, with some changes to accommodate F1 scores rather than accuracy.
> > >
> > > The ID test distribution is the SQuAD NLP dataset (https://rajpurkar.github.io/SQuAD-explorer/) which is a reading comprehension dataset, consisting of questions posed by crowdworkers on a set of Wikipedia articles. The OOD dataset Amazon is similarly a reading comprehension dataset over Amazon product reviews (proposed by Miller et al. in [1].)
> > >
> > > Since we are looking at a reading comprehension task, instead of classification, we look at F1 score instead of accuracy. The **F1 score** looks at the average overlap between the prediction and ground truth answer. Similarly, we define *F1 agreement*, which we define as the average F1 score where the prediction of one of the models in the pair is treated as the ground truth answer. Formally, given models $h_1$ and $h_2$,
> > > $$\mathsf{F1 \ Agr} = \frac{\mathsf{F1}(h_1, h_2) + \mathsf{F1}(h_2, h_1)}{2}$$
> > >
> > > We compute the slope using 99 models from the CodaLab SQuAD board, specifically the models provided in this worksheet https://worksheets.codalab.org/worksheets/0x787751bd802040ffbea9f6ccbd27175f and observe a similar slope in ID vs OOD F1 score and F1 agreement (*though we do note that the $R^2$ values are slightly lower than 0.95 and we will make note of this in the paper*).
> > >
> > > Our prediction algorithm ALine-D achieves a MAE of 0.028 for this set of 99 models. *We will include these full experimental details including the list of models in our next update.*
> > >
> > > As you have suggested, we have also made a change to our wording of the trajectory experiment to reflect the fact that the slope slightly deviates when we use models from the same training trajectory. See line 309.
> > >
> > > All changes are marked in blue. Thank you for your thorough feedback!
> > >
> > > [1] https://arxiv.org/pdf/2004.14444.pdf The Effect of Natural Distribution Shift
> > > on Question Answering Models. Miller et al.

---

### Official Review · Reviewer_7Nem · 2022-07-10

**Rating:** 7
**Confidence:** 4
**Soundness:** 3 good
**Presentation:** 3 good
**Contribution:** 3 good

**Summary:**

This paper introduces a finer-grained version of accuracy-on-the-line called agreement-on-the-line, capable of estimating OOD performance without access to OOD labels. Rather than analyzing the correlation between ID and OOD performance metrics like accuracy-on-the-line, the authors analyze the predictions made by pairs of models, and observe that strong agreement between model predictions on ID data correlates with prediction agreement on OOD data. Moreover, agreement-on-the-line can be used as a test for when accuracy-on-the-line is appropriate to use due to the apparent co-occurrence of the two phenomenon. The main advantages of their method over previous works is not having to rely on assumptions about data shift magnitude, and the ability to aggregate information from many pairs of models. also The authors validate their claims empirically on a variety of datasets, and for several model classes.

**Questions:**

Related to W1, it's not obvious if Proposition 1 is unique to neural networks, or just highly accurate models. Specifically, if the ID and OOD accuracy are strongly correlated across the set of models considered, and the models are also accurate, how do you think minimum accuracy affects agreement correlation? The lower bound on agreement is Acc(M1) - (1 - Acc(M2)), so just by following this lower bound, agreement correlation would be strong.

There is a bit of a contradiction in the claim that "In fact, ALine-D supercedes previous methods even when accuracy-on the-line does not hold". Namely, agreement-on-the-line is introduced as a way of determining when accuracy-on-the-line holds, and then leveraging accuracy-on-the-line to predict OOD performance, so if accuracy-on-the-line does not hold, how can it be the case that ALine-D performs so well when the underlying conditions break down?

**Limitations:**

There is almost no discussion of limitations, but a couple come to mind. For example, the baselines used in Figure 2 such as linear models and SVMs are not appropriate for CIFAR10 classification. Understandably the authors wanted to see if their results hold for classical ML models, but such models should be evaluated on datasets where they can achieve good performance since conclusions drawn from poorly performing models can be misleading. Also, the experiments section does not investigate why agreement-on-the-line is so effective. It would be nice to see if there is something particular about the optimization procedure used, i.e. the inductive bias of SGD, that enables agreement-on-the-line to work when using model checkpoints. Since the premise of the paper is that the observed phenomenon is peculiar to neural networks, the conclusion should at least discuss what role architecture plays.

**Strengths And Weaknesses:**

S1: Agreement between models is a clever surrogate for whether accuracy-on-the-line holds. It is fast and simple to compute.

S2: Results are impressive, and consistent across the several datasets considered, particularly for the Camelyon and iWildCam datasets.

S3: The ability to use multiple checkpoints for a single model rather than having to train many independent models is very appealing. This was something not investigated by the accuracy-on-the-line paper, so it is a novel and intriguing result.

W1: An analysis of agreement vs accuracy would be welcome. For example, what is the biggest possible discrepancy in agreement for two models M1 and M2 that have the same ID accuracy Acc? This could be used to bound the difference in slope between the agreement and accuracy lines, when accuracy is not computable.

W2: The definitions of weak (R^2 < 0.75) and strong (R^2 > 0.95) correlation seem a bit arbitrary. Such definitions vary throughout the literature, though an R^2 of 0.75 is generally considered to be quite strong.

W3: Some of the claims in section 5 are a bit hand-wavy. For example, "Interestingly, the other benchmarks also did not perform very well on these datasets, suggesting that perhaps the success of these prediction methods could also partially be attributed to accuracy-on-the-line" needs some more justification.

---

> ### Author Response · Authors · 2022-08-02
> **1/n Response for Reviewer 7Nem**
>
> Thank you for the detailed feedback! We address the concerns below:
>
> > W1: ...what is the biggest possible discrepancy in agreement for two models M1 and M2 that have the same ID accuracy Acc? …The lower bound on agreement is Acc(M1) - (1 - Acc(M2)), so just by following this lower bound, agreement correlation would be strong.
> - Thank you for this insight. Previous works have shown that agreement between deep models do not satisfy the lower bound. Specifically large-scale studies [1] [2] show that neural networks often make similar mistakes.
> - Our Appendix E looks into this question a bit more carefully. Specifically, Jiang et al. [3] showed that the agreement between two models of the same architecture trained with different random seeds is approximately equal to their average test accuracy 0.5 Acc(M1) + 0.5 Acc(M2), which is higher than the lower bound.
>
> > Related to W1, it's not obvious if Proposition 1 is unique to neural networks, or just highly accurate models. Specifically, if the ID and OOD accuracy are strongly correlated across the set of models considered, and the models are also accurate, how do you think minimum accuracy affects agreement correlation?
> - Agreement-on-the-line is _not_ a phenomena that occurs due to models being highly accurate. Even two models that don’t perform well have agreements that lie on this line.
> - The model collections in our experiments consist of neural networks of widely varying ID/OOD accuracy achieved by training for a small number of training epochs. For example, some of our CIFAR10 models achieve less than 20% accuracy. Even pairs of models that do not perform well observe a strong correlation.
> - To make this more explicit, we've updated Appendix E with a scatter plot of ID vs OOD agreement colored by their average performance. We also provide a plot of agreement vs accuracy.
>
> > Understandably the authors wanted to see if their results hold for classical ML models, but such models should be evaluated on datasets where they can achieve good performance since conclusions drawn from poorly performing models can be misleading.
> - The point of this comparison was to show that our phenomena is not as generalizable as accuracy-on-the-line. Specifically, Miller et al. showed that classical ML models also observe a strong correlation between ID vs OOD accuracy regardless of their performance. However, this is not the case for agreement.
>
> > W2: The definitions of weak (R^2 < 0.75) and strong (R^2 > 0.95) correlation seem a bit arbitrary. Such definitions vary throughout the literature, though an R^2 of 0.75 is generally considered to be quite strong.
> - These thresholds are indeed arbitrary, and simply chosen to discretize the behavior of the linear correlation between ID and OOD accuracy to be strong or weak when formalizing the agreement-on-the-line phenomena for the 8 datasets in the main body. In reality, the behavior is more nuanced and likely follows a gradient  i.e. when the R2 value is higher, the slope/bias of ID vs OOD accuracy and ID vs OOD agreement become closer to each other.  This can be observed in some of the other datasets we observe in Appendix B.
> - We can include this discussion in the final version of our paper.
>
> > W3: Some of the claims in section 5 are a bit hand-wavy. For example, "Interestingly, the other benchmarks also did not perform very well on these datasets, suggesting that perhaps the success of these prediction methods could also partially be attributed to accuracy-on-the-line" ...
> - This was an observation that we made, and we do not have further theoretical or empirical justification. To be more careful, we’ve removed this sentence from the results section.

---

> ### Author Response · Authors · 2022-08-02
> **2/n Response for Reviewer  7Nem**
>
> [2/n]
>
> > There is a bit of a contradiction in the claim that "In fact, ALine-D supercedes previous methods even when accuracy-on the-line does not hold". ...how can it be the case that ALine-D performs so well when the underlying conditions break down?
> - The fact that the method is less reliable when correlations are weak does not contradict the fact that it still outperforms all other methods.
> - This is to say that ALine-D degrades more gracefully than other methods. It would be disingenuous if we say we totally understand the mechanism behind this robustness but we believe it is an exciting direction for future work.
>
> > It would be nice to see if there is something particular about the optimization procedure used, i.e. the inductive bias of SGD...
> - This is a really interesting question! We will leave it to be explored in future works.
>
> > ...the conclusion should at least discuss what role architecture plays.
> - We provide an ablation study in Appendix F studying whether a diversity in architecture is important for observing the agreement-on-the-line phenomena. We conclude that architecture diversity is not relevant to observing agreement-on-the-line.
> - At the moment, it’s largely unclear whether specific components of the model architecture induce agreement-on-the-line. We leave this for future work.
>
> Overall, thank you for recognizing the novelty and strength of our work. Please let us know if there are any further questions or concerns.
>
> [1] https://arxiv.org/abs/2006.07710 The Pitfalls of Simplicity Bias in Neural Networks. Shah et al.
>
> [2] https://arxiv.org/abs/2106.07411 Partial success in closing the gap between
> human and machine vision Geirhos et al.
>
> [3] https://arxiv.org/abs/2106.13799. Assessing Generalization of SGD via Disagreement.
>  Jiang & Nagarajan et al.

---

> ### Author Response · Authors · 2022-08-05
> **Any further suggestions?**
>
> Thank you again for the comprehensive and useful review. We just wanted to check if you had any other questions or suggestions - we would love to get feedback to further improve our work. Specifically, we had additional experiments to clarify any misconceptions about our comparison with classical models and the relationship between accuracy and agreement. Agreement-on-the-line does not strictly rely on the fact that neural networks perform well.
>
> The questions you’ve posed are quite interesting, and we hope to extend upon this work in the future to further understand this phenomenon. That said, we believe our empirical findings alone are quite interesting and immediately useful for the deep learning community and practitioners.  Please let us know if you have any further questions or concerns.

---

> ### Author Response · Authors · 2022-08-09
> **Follow-up**
>
> Thank you again for the suggestions for improving the paper. Since today is the last day of the discussion period, we were wondering if all of your concerns have been addressed. If not, we would be happy to continue the discussion and/or revise the paper.

---

### Official Review · Reviewer_ov7g · 2022-07-11

**Rating:** 8
**Confidence:** 4
**Soundness:** 3 good
**Presentation:** 4 excellent
**Contribution:** 3 good

**Summary:**

This paper identifies and extensively analyzes an interesting phenomena relating the agreement and accuracy of models on in-distribution and out-of-distribution data. In particular, they find that when there is a linear correlation between in-distribution test accuracy and out-of-distribution test accuracy across a set of distinct models trained on this data (a concept discovered in an earlier paper), there is also a linear correlation between the *agreements* of pairs of models trained on that data. This phenomena is identified to be unique to neural networks, and somewhat surprisingly, can be used in the reverse direction as a method of unsupervised estimating out-of-distribution accuracy by measuring agreement - even in cases where in-distribution and out-of-distribution agreement is only roughly linearly correlated.

**Questions:**

   * Is there a statistically significant difference between the slopes and biases of neural networks accuracy-on-the-line vs agreement?

**Limitations:**

The paper does a nice job describing its limitations, both in the amount of models that need to be trained as well as its reliance on the accuracy-on-the-line phenomena. The paper could benefit from further discussion on kinds of domain shift where agreement-on-the-line is guaranteed to not hold, e.g. if there were a label shift.

**Strengths And Weaknesses:**

   * **Originality**
     While other papers have used agreement to determine how well a model performs on unlabeled data, to the best of my knowledge, this is a completely novel phenomena that has been identified comparing agreement-on-the-line with the recently discovered accuracy-on-the-line. It is clear how this work differs from previous contributions, and the analysis shows new insights that were not described in previous work.
   * **Quality**
     The claims are mostly well-substantiated, with extensive experimentation supporting the major claims. The methods used are appropriate, and both the strengths and the weaknesses of the work is remarked on. A possible area of improvement could be in the theory that agreement-on-the-line and acccuracy-on-the-line having the same slope and bias for neural networks but not other models - for example, this could be rigorously tested with a hypothesis test to give further evidence to that claim. Another area of improvement could be increased evaluation with respect to the methods for predicting OOD accuracy - for example, the ALine method that only uses a single model would be useful to have in Table 2, especially if it is in the main body of the paper.
   * **Clarity**
     The paper is very clear, both in its organization as well as its figures. It is clear how to run the experiments and reproduce these results, as well as exactly what the two different algorithms are doing. A minor nitpick would be to remove the citation in the abstract so that it can be read completely stand-alone.
   * **Significance**
     The paper is significant both from a theoretical and a practitioners point of view. From the theoretical viewpoint, it provides further evidence that some domain shifts have very particular properties that create both agreement and accuracy-on-the-line phenomena. It is very unclear what that phenomena is or how one might be able to predict whether it is occuring, but it clearly has some interesting implications. For the practitioner, this method provides a method for unsupervised prediction of performance on OOD data that is significantly better than other methods (especially when agreement-on-the-line holds), which is a very valuable contribution that I expect will be used.

---

> ### Author Response · Authors · 2022-08-02
> **Response 1 for Reviewer ov7g**
>
> Thank you for the feedback and recognizing the strengths of our paper. We are also excited about the important practical implications of agreement-on-the-line for predicting OOD accuracy and we hope our novel empirical findings can encourage further research about the structure of natural distribution shifts. We address your questions below.
>
> > Is there a statistically significant difference between the slopes and biases of neural networks accuracy-on-the-line vs agreement?
>
> Thank you for the feedback. We’ve provide the following experiment to test the statistical significance of the differences in slope.
> We randomly sample 1000 subsets of 10 models from the testbed and compute the corresponding difference in slope $a_{acc} - a_{agr}$. We look at the 95% confidence interval of the distribution of these slope differences for the 6 datasets where we observe strong correlation in the main body:
> - CIFAR10.1 (-0.046, 0.017)
> - CIFAR10.2 (-0.120, -0.030)
> - ImageNetV2 (-0.0720, 0.061)
> - CIFAR10C-Fog(-0.077, 0.053)
> - CIFAR10C-Snow (-0.067, 0.047)
> - FMoW (-0.042, 0.030)
>
> Note that for all datasets except CIFAR10.2, the null hypothesis (difference in slopes is 0) lies within the 95% confidence interval, thus it cannot be rejected.
>
> We will include this experiment in the final version of the paper.
>
> > The paper could benefit from further discussion on kinds of domain shift where agreement-on-the-line is guaranteed to not hold, e.g. if there were a label shift.
>
> Although a more rigorous study is necessary, as preliminary findings, we include a Gaussian noise experiment based on theory from Miller, et al. in Appendix A. Miller, et al., considers a toy model where each class comes from a Gaussian distribution. The conclusions of their theory was that ID vs OOD accuracy is strongly correlated for the following distribution shifts:
> - mean shift
> - the covariance matrices of each class remain the same up to a scaling factor
>
> However, a shift where the covariance matrix changes not just by some multiplicative factor does not observe strong ID vs OOD accuracy. We include an experiment from Miller et al. where the distribution shift consists of adding isotropic Gaussian noise to CIFAR-10, versus adding Gaussian noise with the same covariance as that of each class distribution. We show that ID vs OOD accuracy/agreement is only strongly correlated for the latter case, and not the former.
>
> Please let us know if there are any further questions or concerns.

---

> > ### Comment · Reviewer_ov7g · 2022-08-05
> > **Thank you**
> >
> > Thank you for the response and the additional experiments. I think the statistical significance experiment is a great addition to the paper and helps bolster the claim that neural networks tend to observe the same slope and bias for agreement and accuracy on the line.
> >
> > The discussion on the Gaussian noise experiment makes sense to me, and I appreciate the additional findings.
> >
> > An interesting direction (likely outside the scope of the current paper, so certainly not expecting anything here) would be to determine whether agreement on the line holds for a single model by perturbing the dataset instead - for example, some recent uncertainty estimation works and the self-supervised literature augment the dataset in some way and check to see whether agreement occurs between the augmented and the normal dataset (or simply just consistency in the latent space). I would be curious whether there is a link between these two concepts.
> >
> > I will definitely respond here if I have any other questions or concerns, but wanted to give my initial response promptly.

---

> > > ### Author Response · Authors · 2022-08-06
> > > **Thank you**
> > >
> > > Thank you for getting back!
> > >
> > > You make an interesting point and we agree it could be promising future work to understand what agreement-on-the-line says about the way agreement is used in communities outside of OOD prediction eg. self-supervised learning, domain adaptation, etc. It would be exciting to understand why neural networks display this agreement-on-the-line behavior, and use this to guide future algorithms.

---

### Official Review · Reviewer_7TCt · 2022-07-11

**Rating:** 7
**Confidence:** 4
**Soundness:** 3 good
**Presentation:** 3 good
**Contribution:** 3 good

**Summary:**

This paper describes the agreement-on-the-line phenomenon: instead of relying on accuracy-on-the-line, the paper shows that it is possible to use the agreement of a pair of models as a proxy for the ID vs OOD performance tradeoff. Agreement-on-the-line can be estimated solely from unlabelled data and can be used to predict potential OOD performance. Empirical results based on CIFAR-10, ImageNet, and WILDS OOD data show good OOD performance predictions.

**Questions:**

- Under what circumstances would we expect an OOD data point to have a widespread label agreement across models? Is it enough for accuracy-on-the-line to hold for this to be true? If so, why?
- What happens for $0.75 < R^2 < 0.95$?
- Agreement (equation 2) is defined pairwise between two models. Would it be more appropriate to directly formulate it as a quantity (potentially) involving more than just two models and thereby bypass estimation of individual $A_{ij}$'s and $b_i$'s?
- Can you provide more details as to why Prop (ii) only holds for neural networks? Is this merely an empirical finding or can you provide a theoretical justification?

**Limitations:**

The paper does not appear to explicitly discuss major limitations of the presented work. I was not able to find other limitations explicitly mentioned. It would be great if the authors could comment on the first weakness (and question) listed above.

**Strengths And Weaknesses:**

## Strengths

- Estimating OOD detection and generalization performance of ML models is an important topic.
- The paper is well written and easy to follow.
- The proposed method of considering the disagreement between multiple models is simple and does not require labeled OOD data access.
- Disagreement ensembling can also be performed over checkpointed models instead of multiple fully trained models.

## Weaknesses

- Although it seems unlikely that a broad set of models would all agree on the same prediction for an OOD data point, this event probably has non-zero probability, especially for a low number of models. This means that the proposed method can probably not guarantee to be a good OOD performance predictor under all circumstance. This difficulty seems to stem from the fact that behavior on in-distribution data is well understood while this is not the case for OOD data.
- Baseline methods for experimental section are not well discussed. It would be helpful to introduce these methods in the related works section.
- Limitations are not well discussed.

---

> ### Author Response · Authors · 2022-08-02
> **Response 1 for Reviewer 7TCt**
>
> Thank you for the detailed feedback. We respond to your comments below:
>
> > Although it seems unlikely that a broad set of models would all agree on the same prediction for an OOD data point, this event probably has non-zero probability, especially for a low number of models.
> - We should emphasize that we are not studying point estimates but rather the accuracy over the _population_ of data.  Empirically, we observe that the models do not agree over all OOD test points. In our work, we show that agreement can be used to predict OOD accuracy (aggregated over all the data points).
> - We discuss the concern regarding the number of models in the paper. We have an ablation study in Appendix F which is referred to in the main body in line 320 (We’ve attached the appendix to the main pdf. Previously the appendix was in the supplementary material.) looking at how many models are necessary for ALine-D to make accurate predictions.
> - The conclusions of our ablation study was that for the CIFAR10 related shifts, even less than 10 models was generally sufficient and more models than that did not improve the prediction performance.  For ImageNetV2, the prediction performance improved with more models.
>
> > This means that the proposed method can probably not guarantee to be a good OOD performance predictor under all circumstances. This difficulty seems to stem from the fact that behavior on in-distribution data is well understood while this is not the case for OOD data.
> - We do not claim ALine-D will perform well under all circumstances, but precisely only when agreement-on-the-line holds i.e. ID vs OOD agreement is strongly linearly correlated.
> - We want to clarify any misunderstanding and state that this is _not_ because of the concern raised previously on whether models agree point-wise OOD: we only measure aggregate performance and OOD agreement is not directly equal to OOD accuracy, but rather via a linear system.
> - Furthermore, as stated in our conclusions, we do not claim that agreement-on-the-line is a universal phenomenon, and indeed it is likely that one can construct synthetic examples where it might be violated. However, all empirical evidence points to this phenomenon being very widespread and thus quite practical, but we hope to further understand this empirical phenomena in future research.
>
> > Agreement (equation 2) is defined pairwise between two models. Would it be more appropriate to directly formulate it as a quantity (potentially) involving more than just two models and thereby bypass estimation of individual $A_{ij}$'s and $b_i$'s?
> - This is an interesting point but we are not sure if there is a clear way to extend the definition of agreement to be between the predictions of >=3 models.
> - Given n models, the linear system of equations constructed in ALine-D solves for n unknowns. Using all  n choose 2 pairs may not be necessary if the linear correlation of ID vs OOD accuracy is very strong, but we construct the system using all pairs in our algorithm for simplicity sake
> - The $A_{ij}$’s are not estimated, but set to be constants ½ (i.e., we are simply taking the average).
>
> > Baseline methods for the experimental section are not well discussed. It would be helpful to introduce these methods in the related works section.
> - Thank you for the feedback, we will describe the baseline methods in more detail in the final draft due to the current page limit.
>
> > What happens for 0.75<R2<0.95?
> - These thresholds are indeed arbitrary, and simply chosen to discretize the behavior of the linear correlation between ID and OOD accuracy to be strong or weak when formalizing the agreement-on-the-line phenomena for the 8 datasets in the main body. In reality, the behavior is more nuanced and follows a gradient i.e. when the R2 value is higher, the slope/bias of ID vs OOD accuracy and ID vs OOD agreement become closer to each other. This can be observed in our Appendix B where we look at around 20 other datasets.
> - We can include this discussion in the final version of our paper.
>
> > Can you provide more details as to why Prop (ii) only holds for neural networks? Is this merely an empirical finding or can you provide a theoretical justification?
> - Our paper identifies an empirical phenomenon, though we’ve found this phenomena to be true across hundreds of models. We emphasize this fact in our work in lines 62-66, 331. We hope it inspires interesting theoretical discussion about distribution shifts and the behavior of neural networks in the future.
>
> Above we clarify several potential misunderstandings of our method. Specifically, that we are looking at the agreement over a distribution, not a point. The reviewer was overall concerned about a lack of discussion about the limitations of our prediction method. In our reply above, we point to our ablation study in Appendix F, and lines in our paper that address their specific concerns.
>
> Please let us know if there are any further questions/concerns.

---

> > ### Comment · Reviewer_7TCt · 2022-08-05
> > **Thank you!**
> >
> > I thank the authors for their response! My concerns have been largely addressed and I have raised my score. I would still strongly encourage the authors to:
> >
> > - add a short description of the baseline methods; and
> > - discuss their reasoning behind choosing the $R^2$ thresholds.

---

> > > ### Author Response · Authors · 2022-08-06
> > > **Thank you**
> > >
> > > Thanks a lot! We will make sure to make these changes in the next revision.

---

### Author Response · Authors · 2022-08-05
**Thank you**

Dear reviewers,

Thank you taking the time to provide us careful and thoughtful feedback. As the reviewer’s have unanimously pointed out, our empirical finding that ID vs OOD agreement follows the same linear trend as ID vs OOD accuracy when the latter is strongly correlated is very novel, and has important implications. Our paper conducts an extensive study of this phenomenon agreement-on-the-line across hundreds of models and 20+ OOD datasets.

Several reviewers have asked us to further delve into the limitations of the empirical findings and our consequent prediction method. In our rebuttal, we have pointed to our experiments in the appendix that addresses specific concerns.
We have also updated the paper submission with several additional experiments based on reviewer feedback. Specifically, we have included

- 2 extra experiments in Appendix E answering reviewer’s questions about whether agreement-on-the-line is a consequence of neural networks performing well on the ID/OOD datasets. Our experiments show that the **phenomena holds true even for a set of models that do not perform well**.
- ImageNetv2 trajectory experiment in Section 5.3. We show that **our prediction method can be used to predict the ImageNetv2 performance of checkpoints** collected along the training trajectory of ImageNet.
- Experiment that shows that **there does not exist a statistically significant difference between the ID vs OOD agreement linear fit and the ID vs OOD accuracy linear fit** (under the reply for Reviewer 2).
- **Results on new datasets, including NLP datasets**. We plot the ID vs OOD linear fit of accuracy and agreement for two extra datasets RxRx1-Wilds and Amazon. Amazon is a reading comprehension NLP dataset.

More importantly, our paper is interesting precisely because _our prediction method is coupled with its well characterized limitations_ i.e. ALine-D performs well only when agreement-on-the-line holds which can be empirically verified by checking the linear correlation of ID vs OOD agreement.

Even though why agreement-on-the-line occurs is largely a mystery, we believe that the contribution of this empirical phenomena alone is extremely important and novel to understanding distribution shifts and the behavior of neural networks.

Again, thank you for your time, and we welcome any further comments or feedback to improve upon our work.

---

### Meta-Review · Area_Chair_8mwF · 2022-08-25

**Recommendation:** Accept
**Confidence:** Certain

**Metareview:**

This work addresses the “agreement-on-the-line phenomenon” by extensively analyzing a phenomenon relating to the agreement and accuracy of models on in-distribution and out-of-distribution data. In particular, one of the findings is that when there is a linear correlation between in-distribution test accuracy and out-of-distribution test accuracy across a set of distinct models trained on this data, there is also a linear correlation between the agreements of pairs of models trained on that data. Agreement-on-the-line can be estimated solely from unlabelled data and can be used to predict potential OOD performance. The main advantages of their method over previous works are not having to rely on assumptions about data shift magnitude, and the ability to aggregate information from many pairs of models. The paper proposes both a pair-model-based assessment as well as a multi-model-based assessment which allows for noise and potential bias reductions. Empirical results based on CIFAR-10, ImageNet, and WILDS OOD data show good OOD performance predictions.

The paper convinces in all four categories (originality, quality, clarity, and significance), and the reviewers all agree on accepting this work for publication. For the camera-ready version, it would be great if the authors could include a short description of the baseline methods and briefly discuss the reasoning behind choosing the R2 threshold.

**Award:**

No

---

### Decision · Program_Chairs · 2022-09-14

Accept